# Significant underestimation of peatland permafrost along the Labrador Sea coastline in northern Canada

Yifeng Wang[1], Robert G. Way[1], Jordan Beer[1], Anika Forget[1], Rosamond Tutton[1,2], Meredith C. Purcell[3]

[1]Northern Environmental Geoscience Laboratory, Department of Geography and Planning, Kingston, K7L 3N6, Canada

[2]Global Water Futures, Wilfrid Laurier University, Yellowknife, X1A 2P8, Canada

[3]Torngat Wildlife, Plants, and Fisheries Secretariat, Happy Valley-Goose Bay, A0P 1E0, Canada

*Correspondence to*: Yifeng Wang (yifeng.wang@queensu.ca)

**Abstract.** Northern peatlands cover approximately four million km², and about half of these peatlands are estimated to contain permafrost and periglacial landforms, like palsas and peat plateaus. In northeastern Canada, peatland permafrost is predicted to be concentrated in the western interior of Labrador but is assumed to be largely absent along the Labrador Sea coastline. However, the paucity of observations of peatland permafrost in the interior, coupled with traditional and ongoing use of perennially frozen peatlands along the coast by Labrador Inuit and Innu, suggests a need for re-evaluation of the reliability of existing peatland permafrost distribution estimates for the region. In this study, we develop a multi-stage consensus-based point inventory of peatland permafrost complexes in coastal Labrador and adjacent parts of Quebec using high-resolution satellite imagery, and we validate it with extensive field visits and low-altitude aerial photography and videography. A subset of 2092 wetland complexes that potentially contained peatland permafrost were inventoried, of which 1119 were classified as likely containing peatland permafrost. Likely peatland permafrost complexes were mostly found in lowlands within 22 km of the coastline where mean annual air temperatures often exceed +1 °C. A clear gradient in peatland permafrost distribution exists from the outer coasts, where peatland permafrost is more abundant, to inland peatlands, where permafrost is generally absent. This coastal gradient may be attributed to a combination of climatic and geomorphological influences which lead to lower insolation, thinner snowpacks, and poorly drained, frost-susceptible materials along the coast. The results of this study suggest that existing estimates of permafrost distribution for southeastern Labrador require adjustments to better reflect the abundance of peatland permafrost complexes to the south of the regional sporadic discontinuous permafrost limit. This study constitutes the first dedicated peatland permafrost inventory for Labrador and provides an important baseline for future mapping, modelling, and climate change adaptation strategy development in the region.

## 1 Introduction

Near the southern boundary of latitudinal permafrost zonation, lowland perennially frozen ground is primarily restricted to wetlands in the form of palsas (peat mounds with a frozen core of mineral and organic material) and peat plateaus (fields of frozen peat elevated above the general surface of the surrounding peatland) (Payette, 2004; International Permafrost Association Terminology Working Group, 2005; Zoltai, 1972; Zoltai and Tarnocai, 1975). Persistence of these cryotic

landforms at the extreme limits of their viability is facilitated by a large temperature offset between the ground surface and the top of permafrost, caused by the thermal properties of thick layers of overlying peat and the buffering effect of ground ice (Burn and Smith, 1988; Williams and Smith, 1989). In recent years, many studies have shown that peatland permafrost can be very sensitive to climate warming and ecosystem modifications (Beilman et al., 2001; Borge et al., 2017; Thibault and Payette, 2009). Understanding the distribution of these ice-rich, thaw-sensitive periglacial environments is important for assessing thermokarst potential (Gibson et al., 2021; Olefeldt et al., 2016), local hydrological and vegetation change (Zuidhoff and Kolstrup, 2005), regional infrastructure or land-use planning, and global carbon stores and carbon cycling activities (Hugelius et al., 2014).

Palsas and peat plateaus are primarily thought to occur in continental locations (Fewster et al., 2020; Hustich, 1939) where colder winters allow deeper frost penetration and drier summers promote less thaw. As such, palsas and peat plateaus have been described in many continental locations in Canada, including Yukon Territory, the Northwest Territories, and the Prairie provinces (e.g., Beilman et al., 2001; Coultish and Lewkowicz, 2003; Mamet et al., 2017; Thie, 1974; Zoltai, 1972). However, these landforms have also been documented in coastal locations including the Hudson Bay Lowlands in northern Manitoba, Ontario, and Quebec (e.g., McLaughlin and Webster, 2014; Ou et al., 2016; Pironkova, 2017). In the Labrador region of northeastern Canada, continental- to hemispheric-scale studies have depicted peatland permafrost as present in the region's continental interior but as far less abundant or completely absent along most of the Labrador Sea coastline (Fewster et al., 2020; Hugelius et al., 2020; Olefeldt et al., 2021). However, historic and contemporary use of coastal peatland permafrost environments by Labrador Inuit and Innu is well documented (Anderson et al., 2018), and published field-based observations (e.g., Anderson et al., 2018; Andrews, 1961; Brown, 1975, 1979; Davis et al., 2020; Dionne, 1984; Elias, 1982; Hustich, 1939; Seguin and Dionne, 1992; Smith, 2003; Way et al., 2018; Wenner, 1947) suggest that peatland permafrost is abundant along some sections of the coast. This recurring misestimation of peatland permafrost has an impact on predictions of ground ice content (O'Neill et al., 2019), thermokarst potential (Olefeldt et al., 2016), and carbon content (Hugelius et al., 2014) in the region.

Locally, preservation of peatland permafrost complexes is relevant to Labrador Inuit and Innu because these areas are frequented for traditional activities such as bakeapple (cloudberry; Inuttitut: appik; Innu-aimun: shikuteu; *Rubus chamaemorus*) berry-picking (Anderson et al., 2018; Karst and Turner, 2011; Norton et al., 2021), goose hunting, and fox trapping (Way et al., 2018). Improvements to our understanding of regional peatland permafrost distribution will provide an important baseline for local and regional climate change adaptation strategy development, while better representation of the distribution of thaw-sensitive terrain will inform future development of linear and built infrastructure in coastal Labrador (Way et al., 2021b; Bell et al., 2011).

Previous peatland permafrost mapping in Labrador has been limited to scattered observations of palsa bogs from the National Topographic Database (Natural Resources Canada, 2005) and the Ecological Land Classification (Environment Canada, 1999), with no comprehensive peatland permafrost inventorying efforts completed to date (Way et al., 2018). In this study, we develop a multi-stage, consensus-based point inventory of contemporary peatland permafrost complexes within 100

km of the Labrador Sea coastline (Figure 1), comprising the region of Nunatsiavut and surrounding areas, including the land
claims agreement-in-principle of the Labrador Innu Nation (Nitassinan) and coastal areas claimed by the NunatuKavut
Community Council (NunatuKavut). The goal of this inventory is to map and contextualize the contemporary distribution of
peatland permafrost complexes throughout coastal Labrador, using extensive validation efforts from a combination of field
visits and low-altitude image and video acquisitions. We hypothesize that this point-based inventory will reinforce the local
understanding of a high abundance of peatland permafrost landforms in coastal locations, which will be relevant for carbon
modelling, land use planning, infrastructure development, and climate change adaptation strategy development at local to
regional scales in northeastern Canada. This contribution will also provide insights into the reliability of relevant peatland
permafrost and permafrost distribution products, which currently claim an absence or low abundance of both peatland
permafrost and permafrost along the Labrador Sea coastline. Based on our results, we also propose amendments to the current
limits of the sporadic discontinuous and isolated patches of permafrost distribution zones in southeastern Labrador. This point-
based inventory is a first step towards understanding the distribution of peatland permafrost in Labrador and will contribute to
refined regional and global estimates of ground ice content, thermokarst potential, and carbon storage in northern Canada.

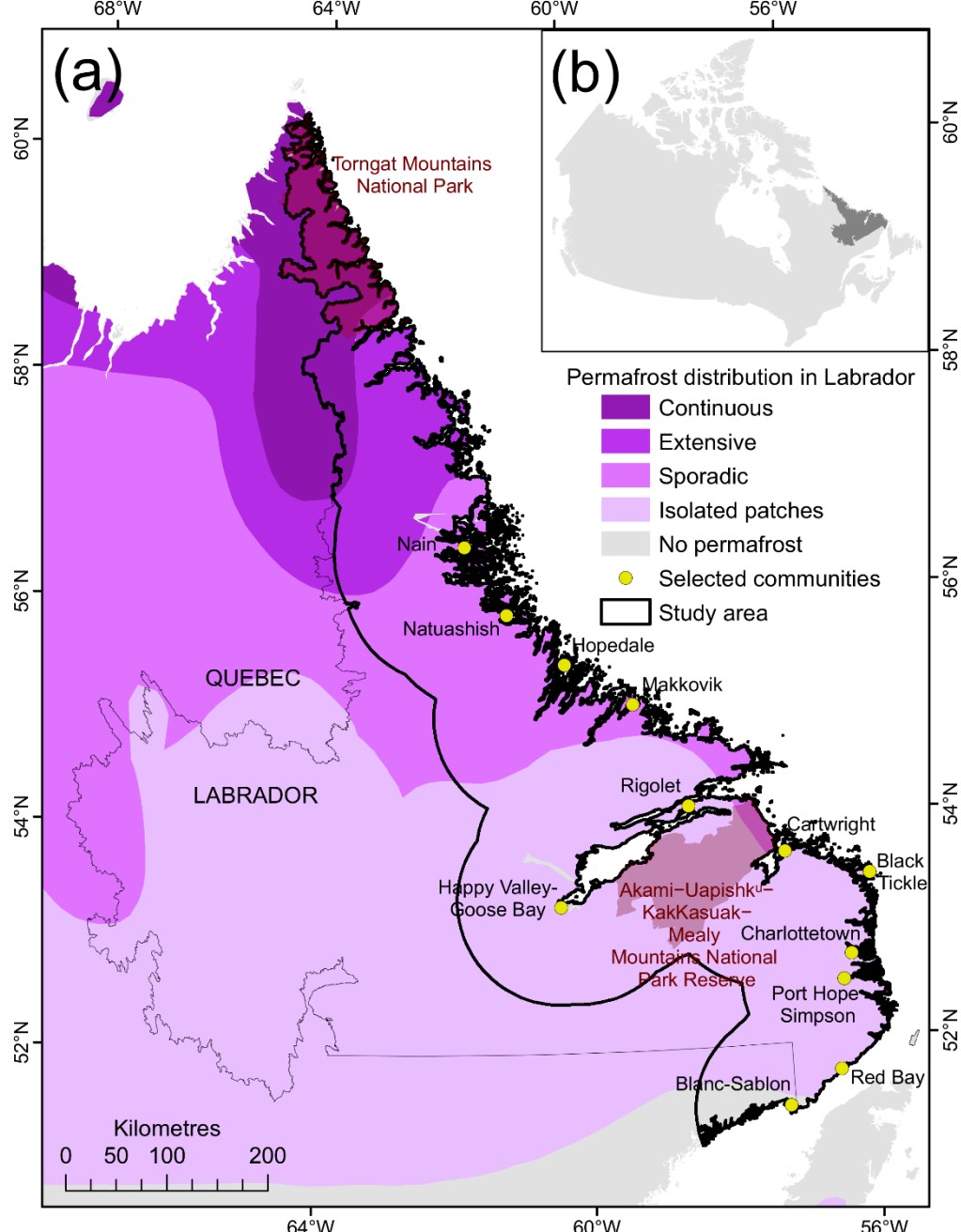

**Figure 1. (a) Permafrost zonation in Labrador (Heginbottom et al., 1995) with the boundary for the inventory study area (black line)**
**corresponding to areas within 100 km of the Labrador Sea coastline. Map is annotated with locations of the Torngat Mountains**
**National Park, Akami−Uapishkᵁ−KakKasuak−Mealy Mountains National Park Reserve, and selected communities; (b) Inset map**
**showing Labrador's position in Canada.**

## 2 Study area

### 2.1 Bioclimatic setting

Labrador's climate is strongly influenced by atmosphere-ocean interactions from the adjacent Labrador Sea (Barrette et al., 2020; Way and Viau, 2015). In coastal Labrador, long, cold winters and short, cool summers are largely dictated by the Labrador Current that carries cold Arctic waters down the eastern coast of mainland Canada (Banfield and Jacobs, 1998; Foster, 1983; Roberts et al., 2006; Way et al., 2017). Mean annual air temperatures (1980-2010) decrease with continentality and latitude, ranging from -12°C in parts of the Torngat Mountains National Park to +1.5°C near the community of Blanc-Sablon (Karger et al., 2017, 2021). Labrador is also characterized by some of the highest precipitation amounts in the North American boreal zone (Banfield and Jacobs, 1998; Hare, 1950) due to its varying relief, high moisture availability from the adjacent Atlantic Ocean, and high frequency of passing winter storm systems (Brown and Lemay, 2012). Precipitation totals as high as ~2700 mm per year are estimated for some locations at high elevations along the coast (Karger et al., 2017, 2021), with solid precipitation fractions increasing with both latitude and elevation (~0.35 at Blanc-Sablon; ~0.5 at Nain) (Environment and Climate Change Canada, 2022).

Ecologically, Labrador is characterized by taiga forests in the interior, tundra in the north, and wind-swept coastal barrens along the coastline of the Labrador Sea (Roberts et al., 2006). Tree cover is sparse in the coastal barrens because of climatic and physiographic limitations, but dense patches of black spruce (*Picea mariana*), white spruce (*Picea glauca*), tamarack (*Larix laricina*), and balsam fir (*Abies balsamea*), interspersed with deciduous trees, like paper birch (*Betula papyrifera*) and trembling aspen (*Populus tremuloides*), exist in sheltered locations and on some slopes (Roberts et al., 2006). Wetlands are found throughout Labrador, but total wetland abundance is difficult to assess given widespread disagreement between existing estimates of wetland and peatland extents for this region (Supplement Sect. S1). Generally, wetlands in Labrador tend to decrease in abundance but increase in size as latitude increases. Most wetlands along the southern Labrador coast are classified as raised bogs, while inland, most wetlands are string and blanket bogs (Foster and Glaser, 1986).

### 2.2 Physical environment

Labrador is mostly underlain by igneous and metamorphic bedrock (Roberts et al., 2006). Extensive blankets of glacial till were deposited during the retreat of the Laurentide Ice Sheet (12-6 k years BP) (Bell et al., 2011; Dyke, 2004), along with thin layers of medium- to fine-grained marine and glaciomarine sediments in coastal lowland areas below the marine limit (Fulton, 1995). The post-glacial marine limit decreases with latitude, from ~150 m a.s.l. in southeastern Labrador and along the Quebec Lower North Shore to 0 m a.s.l. at the northernmost tip of Labrador in the Torngat Mountains (Dyke et al., 2005; Occhietti et al., 2011; Vacchi et al., 2018). The broad distribution of near-surface bedrock and hardpans (Smith, 2003) results in poor drainage that has facilitated peatland development across large areas of southern Labrador, particularly in depressions and over flat deposits.

## 2.3 Permafrost distribution

While permafrost conditions in Labrador, including the presence of peatland permafrost landforms, have been noted during ecological, palynological, glaciological, and archeological surveys and studies (Anderson et al., 2018; Andrews, 1961; Hustich, 1939; Smith, 2003; Wenner, 1947), permafrost-specific field investigations are limited to R.J.E. Brown's (1975) helicopter survey in the late 1960s and the Labrador Permafrost Project that began in 2013 (Way, 2017). Our understanding of permafrost distribution in Labrador has relied on extensive extrapolation of limited field observations and broad assumptions of the interactions between air temperature, vegetation cover, snow cover, and permafrost presence (Ives, 1979). According to the Permafrost Map of Canada (Heginbottom et al., 1995), the area underlain by permafrost in Labrador is less extensive than comparable regions in northern Canada like Yukon Territory or the Northwest Territories. Approximately two-thirds of Labrador is classified in the isolated patches of permafrost zone (<10 % permafrost by area), but the distribution of permafrost does become more widespread farther north (Figure 1). Along the Labrador coastline, the sporadic discontinuous permafrost zone (10-50 % permafrost by area) extends slightly further south along the outer edge of the Akami−Uapishkᵁ−KakKasuak−Mealy Mountains National Park Reserve than in the interior, though the justification for this departure is not clarified in published literature. Continuous permafrost (>90 % permafrost by area) is expected to persist only at high elevations and latitudes, mostly in the Torngat Mountains (Heginbottom et al., 1995).

## 2.4 Inventory extent

This study is focused on the coastal areas of Labrador and Quebec, within 100 km of the Labrador Sea coastline (Figure 1). This area of interest was informed by knowledge gained from prior works in the region (Anderson et al., 2018; Andrews, 1961; Brown, 1975, 1979; Davis et al., 2020; Dionne, 1984; Elias, 1982; Hustich, 1939; Seguin and Dionne, 1992; Smith, 2003; Way, 2017; Way et al., 2018; Wenner, 1947) that indicated a greater abundance of peatland permafrost landforms along the coast as compared to the interior of Labrador. Exhaustive descriptions of records of peatland permafrost and other periglacial landforms in Labrador have been presented by Brown (1979) and Way (2017), both of whom found limited evidence of peatland permafrost in Labrador's interior.

## 3 Methods

Palsas and peat plateaus are typically found in bogs and may measure up to 4 m higher than their surrounding wetlands, so large peatland permafrost landforms can be identified and mapped from high-resolution satellite imagery (Borge et al., 2017; Gibson et al., 2020, 2021). Our point inventory, which includes only the largest and most visually apparent peatland permafrost complexes within 100 km of the Labrador Sea coastline, was generated through a multi-stage mapping and consensus-based review process, supported by extensive validation efforts mostly completed between 2017 and 2022. Mapping and identification activities were informed by existing wetland and peatland distribution products (Supplement Sect. S1), but

significant disagreement between these products limited their direct application and utility during the inventorying process.
An initial inventory of wetlands of interest (WOIs) was developed as a subset of the wetlands in coastal Labrador deemed
potentially suitable (e.g., bogs and fens) for the development and persistence of peatland permafrost landforms. The presence
of peatland permafrost landforms within the WOIs was then evaluated through a consensus-based review of high-resolution
satellite imagery by three mappers with permafrost-specific field experience in the region. Final interpretation of peatland
permafrost presence or absence within the WOIs was based on reviewer agreement and was informed by field- and imagery-
based validation of peatland permafrost landform presence or absence.

## 3.1 Data sources

WOIs were identified and evaluated using Maxar (Vivid) optical satellite imagery, available as the World Imagery
basemap via ArcGIS Online (0.5 m ground sampling distance; 5 m absolute spatial accuracy) (Esri, 2022). These satellite
imagery mosaics consisted of summer imagery with minimal cloud and snow cover, with acquisition dates for Labrador
ranging from 2010 to 2020.
Topographic data from Natural Resources Canada covering the WOIs were extracted from the Canadian Digital
Elevation Data (CDED; 50 m spatial resolution), with a small gap near the provincial border between Labrador and the Quebec
Lower North Shore that was filled in using the Canadian Digital Surface Model (CDSM). Gridded mean annual air temperature
(MAAT) and mean annual thawing degree days (TDD) for the 1981 to 2010 climate normal were extracted from CHELSA
V2.1 (~1 km spatial resolution) (Karger et al., 2017, 2021) at the WOI locations. Mean annual freezing degree days (FDD) for
the WOI locations for 1981 to 2010 were calculated from MAAT and TDD over the same climate normal, following prior
work in the region (Way et al., 2017; Way and Lewkowicz, 2018).

## 3.2 Inventorying peatland permafrost complexes

### 3.2.1 Identifying wetlands of interest (WOIs)

A team of three mappers used ArcGIS Online to identify and place point features within WOIs throughout coastal
Labrador (Figure 2). The point-based nature of the inventorying process allowed for evaluation of the entire study area by
incorporating field- and imagery-based validation for many WOIs over a large study area, as opposed to detailed validation of
peatland permafrost areal coverage within a given WOI. These WOIs were restricted to include only those that contained
prospective peatland permafrost landforms exceeding 2 m in length or width (~4 m$^2$), which was determined to be the smallest
detectable feature based on the 0.5 m spatial resolution of the satellite imagery. Mappers were instructed to identify WOIs
based on local geomorphology, local hydrology and drainage patterns, the presence of a white or grey lichen surface cover
corresponding to *Cladonia* and/or *Ochrolechia* spp. lichens, shadows indicative of elevated landform edges and surface uplift,
and thermokarst ponding or exposed peat indicative of thaw processes. The inventory sought to only include contemporary
peatland permafrost landforms, so WOIs with extensive thermokarst ponding but no evident peatland permafrost landforms
were not included in the database. Individual WOIs ranged in size from ~0.2 km² to larger than ~3.5 km². However, the total
area underlain by peatland permafrost within each WOI was not able to be reliably evaluated using satellite imagery. WOIs
near one another were sometimes difficult to discern due to potential connectivity between adjacent systems, but contiguous
WOIs could generally be identified by differences in drainage, vegetation, and morphology, or because of separation by linear
infrastructure like roads, airstrips, and trails (Figure 2). Mappers also assigned each WOI a self-assessed score to reflect their
confidence in their interpretation of permafrost presence within the wetland complex (1 = low confidence, 2 = medium
confidence, 3 = high confidence).

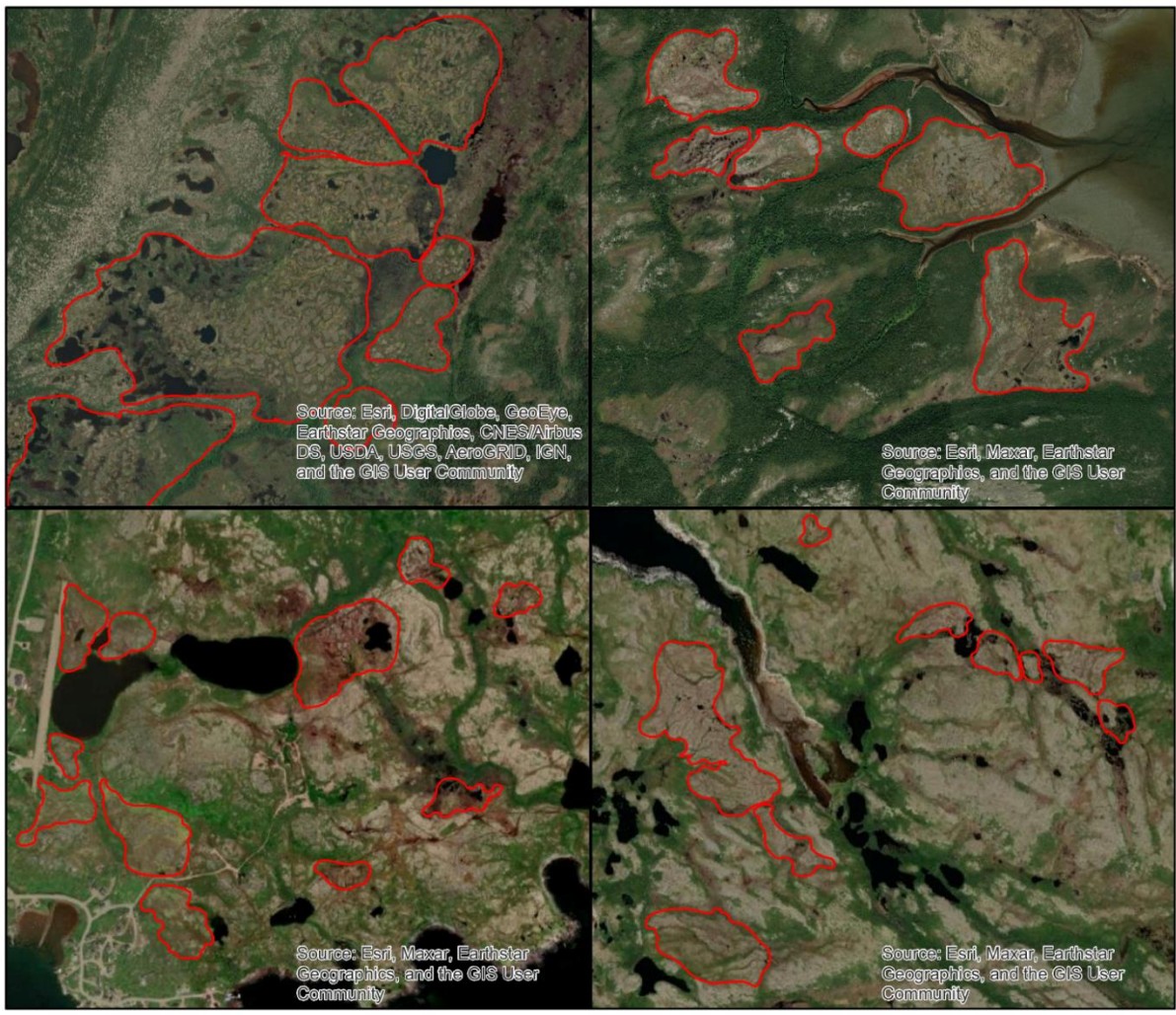

**Figure 2. Examples of wetland complexes of interest (WOIs) in Labrador that were identified by the mapping team using high-**
**resolution satellite imagery available via Esri ArcGIS Online. Examples of WOI boundaries are shown in red and were determined**
**based on differences in drainage or vegetation from adjacent WOIs or based on separation following linear infrastructure, such as**
**roads, airstrips, or trails. Identification was restricted to WOIs that contained prospective peatland permafrost landforms.**

### 3.2.2 Quality control of WOI database

The WOI inventory was subjected to a quality control check, during which complexes were reviewed and duplicates or points clearly not corresponding to wetlands were removed. In some cases, non-wetland locations may have been retained because of difficulties discerning peat plateaus from surface peat over bedrock or coastal tundra.

### 3.2.3 Consensus-based review of WOI database

The quality-controlled WOI inventory was sent back to the mappers for a consensus-based review, similar to Way et al.'s (2021a) approach for rock glacier inventorying in northern Labrador. Each WOI was independently reviewed by two team members, both of whom had access to the mapper's initial confidence rating, and one of whom had access to a field-validated dataset of WOIs (see Sect. 3.3 Validation of subset of WOI database). Both team members were asked to indicate whether each WOI contained peatland permafrost landforms. WOIs evaluated by both reviewers as containing peatland permafrost were considered likely to contain palsas or peat plateaus, while WOIs evaluated by both reviewers as not containing peatland permafrost were considered unlikely to contain palsas or peat plateaus. WOIs with conflicting classifications were considered to possibly contain palsas or peat plateaus. This consensus-based review process resulted in a full inventory of WOIs that were classified as likely, possibly, or unlikely to contain peatland permafrost.

### 3.3 Validation of subset of WOI database

The full, consensus-based inventory results were compared with a field- and imagery-validated dataset of 557 WOIs, with and without contemporary peatland permafrost landforms. From July to September 2021 and 2022, field evaluations of WOIs were undertaken via in-person field visits, remotely piloted aircraft (RPA) image acquisitions (DJI Mini 2 microdrone, weighing less than 250 g), video clip acquisition from a helicopter survey, and image acquisitions from commercial Twin Otter aircraft flights. Interpretation of the presence or absence of permafrost landforms within each WOI that was visited or aerially surveyed was also determined through consensus between two mappers. Any WOIs with disagreements in interpretation were re-evaluated and discussed until consensus could be reached between the two mappers.

Field visits to WOIs were undertaken at road-accessible locations within 500 m of the Trans-Labrador Highway and other accessible side roads via truck or ATV and at coastal locations via speedboat from the nearby communities of Black Tickle, Cartwright, Rigolet, and Nain. The number of WOIs that could be visited for field validation was restricted by weather conditions, tides, the availability of local guides and boat drivers with location-specific expertise, and other logistical and operational constraints. During field visits, team members probed the soil to the depth of refusal (maximum of 125 cm). The nature of refusal, interpreted as frozen ground, compact sediment, clasts, rock, or not applicable (N/A; >125 cm), was noted and used to assess permafrost presence or absence. Where the cause of probe refusal was unclear, instantaneous ground temperature measurements were collected using vertically arranged thermistors connected to an Onset Hobo UX120-006M 4-Channel Analog Data Logger (accuracy ±0.15 °C) (Davis et al., 2020; Holloway and Lewkowicz, 2020; Way et al., 2021b;

Way and Lewkowicz, 2015). Ground temperatures were recorded within the probed hole for a minimum of 10 minutes to allow
for thermal equilibration. Frost probing and instantaneous ground temperature measurements were targeted towards locations
considered most likely to contain frozen ground and thus mostly occurred on elevated peat-covered microtopography within
each WOI.

Low-altitude RPA imagery of prospective peatland permafrost complexes were collected using a DJI Mini 2
microdrone when weather conditions were suitable (i.e., no rain, no fog, low wind). Low-altitude georeferenced video footage
was collected using a GoPro Hero9 camera mounted onto a helicopter during a fuel cache mission in northern Labrador in July
and August 2021, led by the Torngat Wildlife, Plants, and Fisheries Secretariat. The camera was set to record real-time video
(1080 p, 60 fps, wide) at an oblique angle (~45°). The flight altitude was between 90 m and 120 m a.g.l., similar to coastal
Nunavik transects performed by Boisson and Allard (2018), and the flight plan between the Goose Bay Airport and the Torngat
Mountains National Park was designed to fly over WOIs in coastal locations north of the community of Makkovik (55.0° N)
(Supplement Sect. S2). Low-altitude georeferenced aerial images were also collected using handheld digital cameras (Nikon
Coolpix W300 or Olympus Tough TG-6) during commercial Air Borealis Twin Otter flight segments between Cartwright and
Black Tickle and between Goose Bay, Rigolet, Makkovik, Postville, Hopedale, Natuashish, and Nain. The Twin Otter flights
only crossed over WOIs along existing commercial flight routes.

**3.4 Compilation of final WOI database**

The final WOI database of likely, possible, or unlikely peatland permafrost complexes was developed following the
incorporation of the field-validated dataset. WOIs that were classified as likely or possibly to contain peatland permafrost were
subject to a final round of review in which the peatland permafrost landforms were identified as palsas, peat plateaus, or both
palsas and peat plateaus (mixed).

**3.5 Statistical analyses of final WOI database**

ANOVA (analysis of variance) and *post hoc* Tukey's HSD (honest significant difference) tests were performed to
determine whether the mean latitude, distance from coastline, elevation, MAAT, TDD, and FDD were statistically significantly
different between the final classes of likely, possibly, and unlikely peatland permafrost complexes. Statistical analyses were
performed in R 4.0.3 (R Core Team, 2020).

**4 Results**

**4.1 Peatland permafrost complex identification and review**

A total of 2092 unique WOIs, limited to the largest and most visually apparent prospective peatland permafrost
complexes within the study area, were included in the full inventory. Reviewer agreement was very high (89 %) during the
consensus-building review process, with 1116 complexes classified by both reviewers as likely containing peatland permafrost
and 750 complexes classified by both reviewers as unlikely to contain peatland permafrost, and only 226 complexes with
conflicting classifications of permafrost presence or absence (11 %) (Supplement Sect. S2).

**4.2 Validation of peatland permafrost complexes**

In Summer 2021 and 2022, in-person field visits (n=63 WOIs), RPA visits (n=141 WOIs), helicopter video clips

(n=69 WOIs), and Twin Otter images (n=314 WOIs) were combined to evaluate peatland permafrost presence at 531 WOIs,
49 of which were cross-validated using multiple methods (Figure 3; Supplement Sect. S2). Previous work from 2017 to 2020,
including field visits (n=23 WOIs) and RPA image collection (n=19 WOIs), were also used to validate palsa or peat plateau
presence at an additional 19 complexes and peatland permafrost absence at an additional seven complexes (Anderson et al.,
2018; Way, 2017). Out of the 557 WOIs evaluated via field and/or imagery validation methods, 311 were interpreted to contain
peatland permafrost landforms. Comparison between the validation dataset and the consensus-based inventory resulted in re-
classification of 39 of the 226 possible peatland permafrost complexes (17 %) to either likely (n=3) or unlikely (n=36) peatland
permafrost complexes.

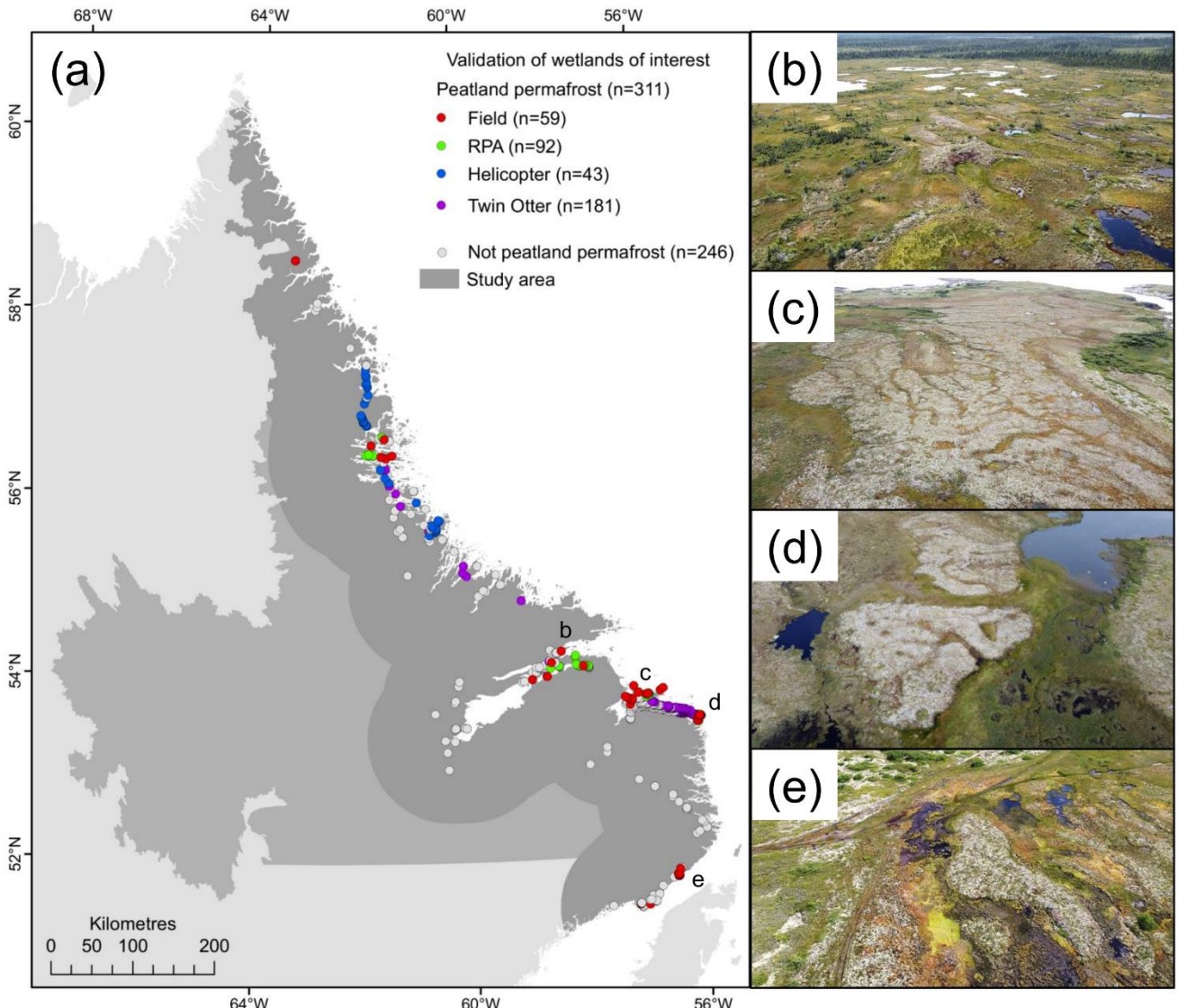

**Figure 3. (a) Locations of validated peatland permafrost complexes in coastal Labrador from field-based activities and imagery**
**acquisition using RPA, helicopter, and Twin Otter from 2017 to 2022; Example of peatland complexes containing palsas and/or peat**
**plateaus near (b) Rigolet, (c) Cartwright, (d) Black Tickle-Domino, and (e) Red Bay.**
**4.3 Peatland permafrost complex inventory**

A total of 1119 out of 2092 WOIs were classified as likely containing peatland permafrost landforms, with an
additional 187 wetland complexes classified as possibly containing peatland permafrost landforms (Figure 4). The largest
clusters of likely and possible peatland permafrost complexes were located between Makkovik (55.0° N) and Black Tickle
(53.5° N) (Figure 4; Figure 5A). The likely peatland permafrost complexes were at low elevation (mean elevation of 29 m
a.s.l.) (Figure 5C) within 22 km of the coastline (mean distance from coastline of 2.6 km) (Figure 4; Figure 5B). Likely peatland
permafrost complexes were distributed from 51.4° N near Blanc-Sablon to 58.6° N in the Torngat Mountains National Park
(Figure 4; Figure 5A), with most complexes located in southeastern Labrador (mean latitude of 54.1° N) (Supplement Sect.
S3). Comparison against gridded climate products showed that the MAAT at peatland permafrost complexes ranged from -7.5
°C to +1.2 °C, with corresponding ranges for FDDs of 1126 degree days to 3471 degree days and TDDs of 733 degree days to
1704 degree days (Figure 5D-F). Despite the wide range in MAAT, the majority of the likely peatland permafrost complexes
(90 %) were found in locations with MAATs between -2 °C and +1 °C (Figure 5D).

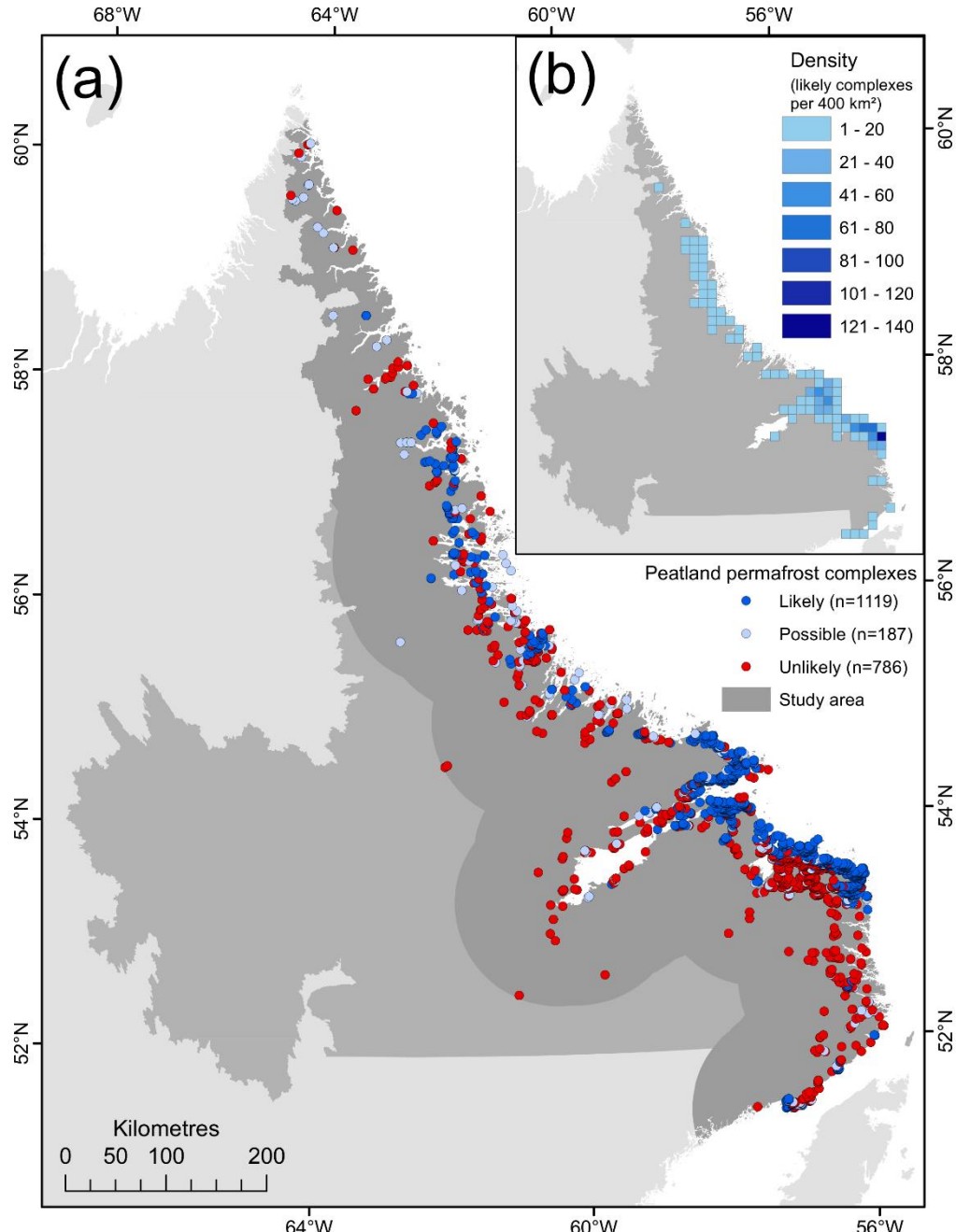

**Figure 4. (a) Spatial distribution of inventoried peatland complexes (n=2092) classified as likely containing peatland permafrost**
**landforms (n=1119), possibly containing peatland permafrost landforms (n=187), and unlikely to contain peatland permafrost**
**landforms (n=786); (b) Inset map showing density of peatland permafrost complexes within 20 by 20 km (400 km²) grid cells.**

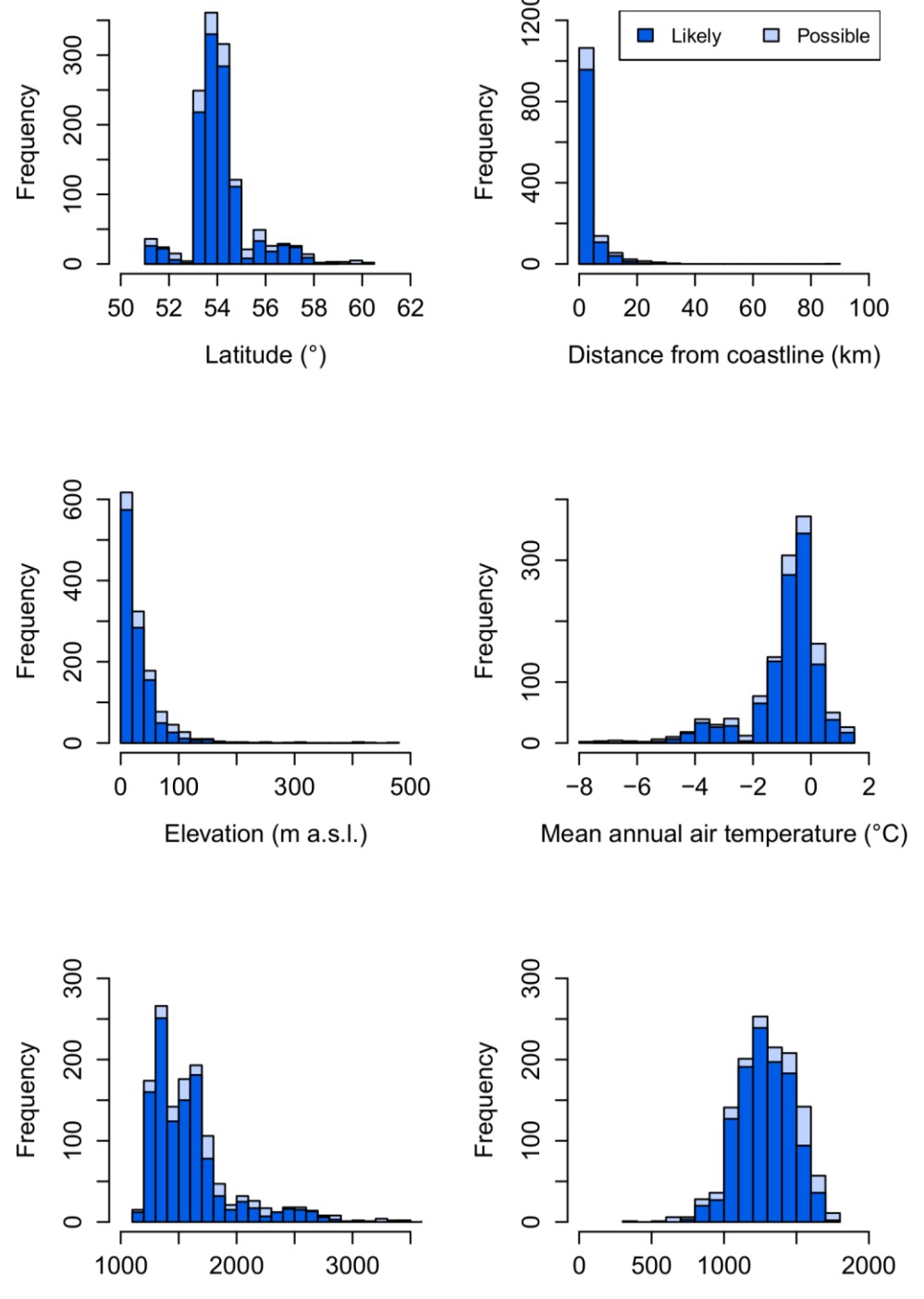

**Figure 5. Distribution of wetland complexes likely or possibly containing peatland permafrost landforms by (a) latitude; (b) distance**
**from the coastline; (c) elevation; and (d) mean annual air temperature; (e) mean annual freezing degree days; and (f) mean annual**
**thawing degree days for the 1981 to 2010 climate normal.**

ANOVA and *post hoc* Tukey's HSD tests revealed that the mean distance from coastline, elevation, MAAT, FDD, and TDD were statistically different between the likely, possibly, and unlikely peatland permafrost complexes at the 95 % confidence level. When compared with the complexes that likely contained peatland permafrost, the 187 complexes that possibly contained peatland permafrost were similarly distributed all along the coastline but were skewed further north (mean latitude of 54.5° N) and extended as far as 60.2° N (Supplement Sect. S3). These less certain features were at greater distances from the coastline (mean distance from coast of 7.8 km) and at higher elevations (mean elevation of 66 m a.s.l.). The 786 complexes that were unlikely to contain peatland permafrost were well distributed between 51.4° N and 60.2° N (Figure 4) but were located further from the coastline (mean distance from coastline of 10.7 km), at higher elevations (mean elevation of 78 m a.s.l.), and at higher MAATs (mean MAAT of -0.5 °C) than the complexes that likely or possibly contained peatland permafrost (Supplement Sect. S3).

Likely and possible peatland permafrost complexes were also classified according to the type of peatland permafrost landforms found within the wetland complex. Complexes that were exclusively comprised of palsas accounted for half of the likely and possible peatland permafrost complexes (50 %) and were distributed along the entire study area. Complexes with exclusively peat plateaus were less common (29 %) and were spatially concentrated between ~53° N and ~55° N. The remaining 21 % of the likely and possible peatland permafrost complexes were interpreted to contain a combination of palsas and peat plateaus, but it is possible that many of these complexes contain dissected and heavily degraded peat plateaus that now resemble palsas. Further field-based investigations would be required to differentiate these degradational landforms.

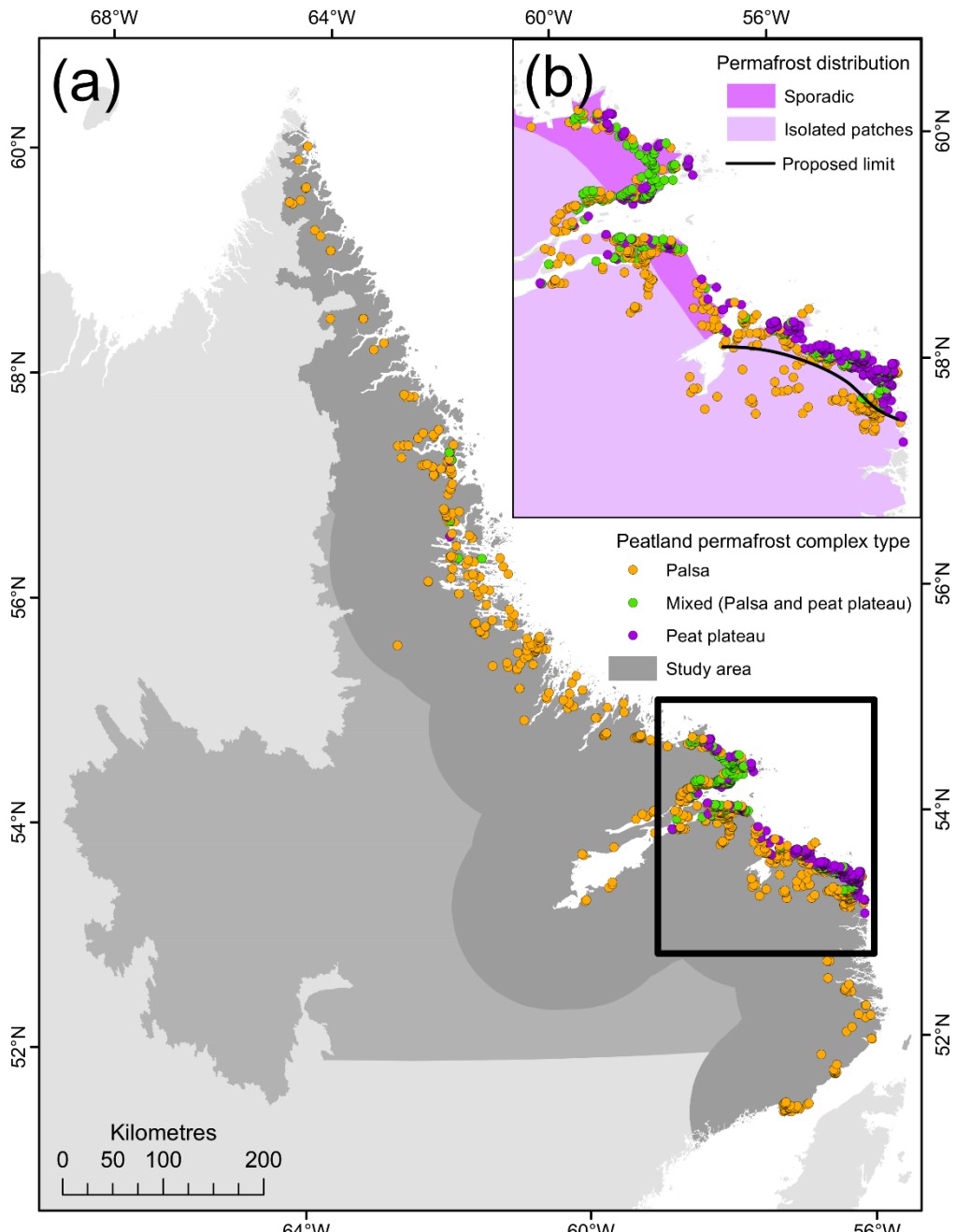

**Figure 6. (a) Spatial distribution of likely and possible peatland permafrost complexes classified by peatland permafrost landform type as palsas, peat plateaus, or a mix of both palsas and peat plateaus for coastal Labrador. (b) Inset map showing existing permafrost distribution zones (Heginbottom et al., 1995) for a subsection of coastal Labrador and the location of a new proposed location for the southern limit of the sporadic discontinuous permafrost zone.**

## 5 Discussion

### 5.1 Distribution of peatland permafrost in Labrador

Peatland permafrost complexes in Labrador and adjacent portions of Quebec are abundant in lowlands within 22 km of the Labrador Sea coastline (Figure 5B). A geographic gradient is especially apparent between Rigolet (54.2° N) and Black Tickle (53.5° N), where peat plateaus are abundant along the coast but absent from wetlands farther inland (Figure 4). The higher density of peatland permafrost complexes along the coast could be linked to climatic factors like persistent fog and cloud cover leading to less incoming solar radiation (Way et al., 2018) or thinner and denser snowpacks (Seppälä, 1994; Vallée and Payette, 2007) in the wind-exposed barrens along the coast (Way et al., 2018). Further work should focus on exploring the role of local climate conditions in the formation and persistence of peatland permafrost in coastal Labrador and similar northern coastal locations. Peatland permafrost was found across a large range of MAATs, spanning from -7.5 °C to +1.2 °C. Permafrost persistence at MAATs above +1 °C in southeastern Labrador was previously noted in a field study at five palsa complexes (Way et al., 2018). Peatland permafrost complexes in Labrador were located at higher MAATs than is predicted for other northern coastal regions like northern Finland, Norway, and Sweden (approximately +0.4 °C) (Parviainen and Luoto, 2007). Our results also suggest that the MAAT threshold of +0.2 °C for peatland permafrost areas previously applied to North America (Fewster et al., 2020) is too low for Labrador and adjacent parts of Quebec where peatland permafrost landforms continue to persist due to their relict and resilient nature (Dionne, 1984; Way et al., 2018). Large thermal offsets (up to and often exceeding 2.0 °C in southeastern Labrador) (Way and Lewkowicz, 2018) are typical of organic-rich landscapes like peatlands and may promote continued landform persistence despite a warming climate (Jorgenson et al., 2010). This may further exacerbate discrepancies between peatland permafrost observations and regional estimates, calling into question the utility of simplified threshold-based approaches when modelling with future climate scenarios. Information on the timing of peatland initiation following deglaciation (Gorham et al., 2007), rates of peat deposition (Tarnocai, 2009; Gorham, 1991), and corresponding peat thicknesses should also be considered in studies of peatland permafrost distribution, as thicker peat deposits may influence permafrost development and protect permafrost persistence through a larger thermal offset (Smith and Riseborough, 2002).

The regional distribution of fine-grained sediments and local depositional history are expected to play an important role in landscape suitability for peatland permafrost landforms (O'Neill et al., 2019; Seppälä, 1986; Zoltai, 1972). For example, differences in the distribution of palsa versus peat plateau landforms have previously been attributed to varying thicknesses and extents of the underlying sediment, with thicker sediment deposits leading to the development of palsas and thinner sediment deposits linked to the development of peat plateaus (Allard and Rousseau, 1999). Differences in sediment grain size may also influence the thickness of the ice lenses and the depth at which they form, with thicker ice lenses developing deeper in finer sediments and thinner ice lenses forming at shallower depths in coarser sediments (Allard and Rousseau, 1999). Further examination of how these variables could influence peatland permafrost formation and persistence in coastal Labrador is challenged by the paucity of information on surficial materials and marine limits along most of the Labrador Sea coastline (Hagedorn, 2022; Occhietti et al., 2011). To date, local marine limits have been identified at some individual locations and

study sites in coastal Labrador (e.g., Bell et al., 2011; Dyke et al., 2005; Occhietti et al., 2011; Vacchi et al., 2018), but widespread mapping of marine sediments has only been completed for a small section of northern coastal Labrador from Goose Bay to Hopedale (Hagedorn, 2022). Based on the information that is currently available, we can qualitatively link the distribution of the largest clusters of peatland permafrost complexes, particularly peat plateau complexes, to locations where post-glacial marine invasions had occurred, such as along the lowland-dominated coastline between Makkovik (55.0° N) and Black Tickle (53.5° N), where frost-susceptible, glaciomarine surficial materials are generally widespread (Fulton, 1989, 1995; Hagedorn, 2022; Occhietti et al., 2011). Meanwhile, fewer peatland permafrost complexes were mapped between Makkovik (55.0° N) and Hopedale (55.5° N), where the elevated topography resulted in limited marine invasions and post-glacial marine deposition along the coast. Significant and coordinated advances in surficial mapping will be required before similar links between peatland permafrost distribution and surficial material type, sediment grain size, and elevation relative to the marine limit can be made for other parts of the coastline.

## 5.2 Implications for peatland permafrost and permafrost distribution in northeastern Canada

Comparisons between our inventory results and several recent national to global wetland, peatland (Supplement Sect. S1; Supplement Sect. S4), and peatland permafrost distribution products (e.g., Fewster et al., 2020; Hugelius et al., 2020; Olefeldt et al., 2021) (Figure 7) provide compelling evidence that peatland permafrost along the Labrador coast is poorly represented by existing datasets. While differences in scale may explain some of this discrepancy, the general pattern presented in most previous datasets, showing relatively greater peatland permafrost in the continental interior and less along the coast, is directly contradicted by the results of this study. This reversed pattern could reflect inaccurate assumptions on the climate limits of peatland permafrost and/or may reflect the absence of field data from many northern coastal peatland permafrost environments (Borge et al., 2017). Inclusion of physiographic variables, like soil conditions, frost-susceptibility of sediments, and more detailed surficial deposit maps are likely necessary for an improved representation of peatland permafrost in northern coastal regions. Recent work by O'Neill et al. (2019), for example, has demonstrated that segregated ice can be reliably modelled along sections of the Labrador Sea coastline (Figure 7D) by incorporating paleogeographic variables like vegetation cover, surficial geology, and glacial lake and marine limits.

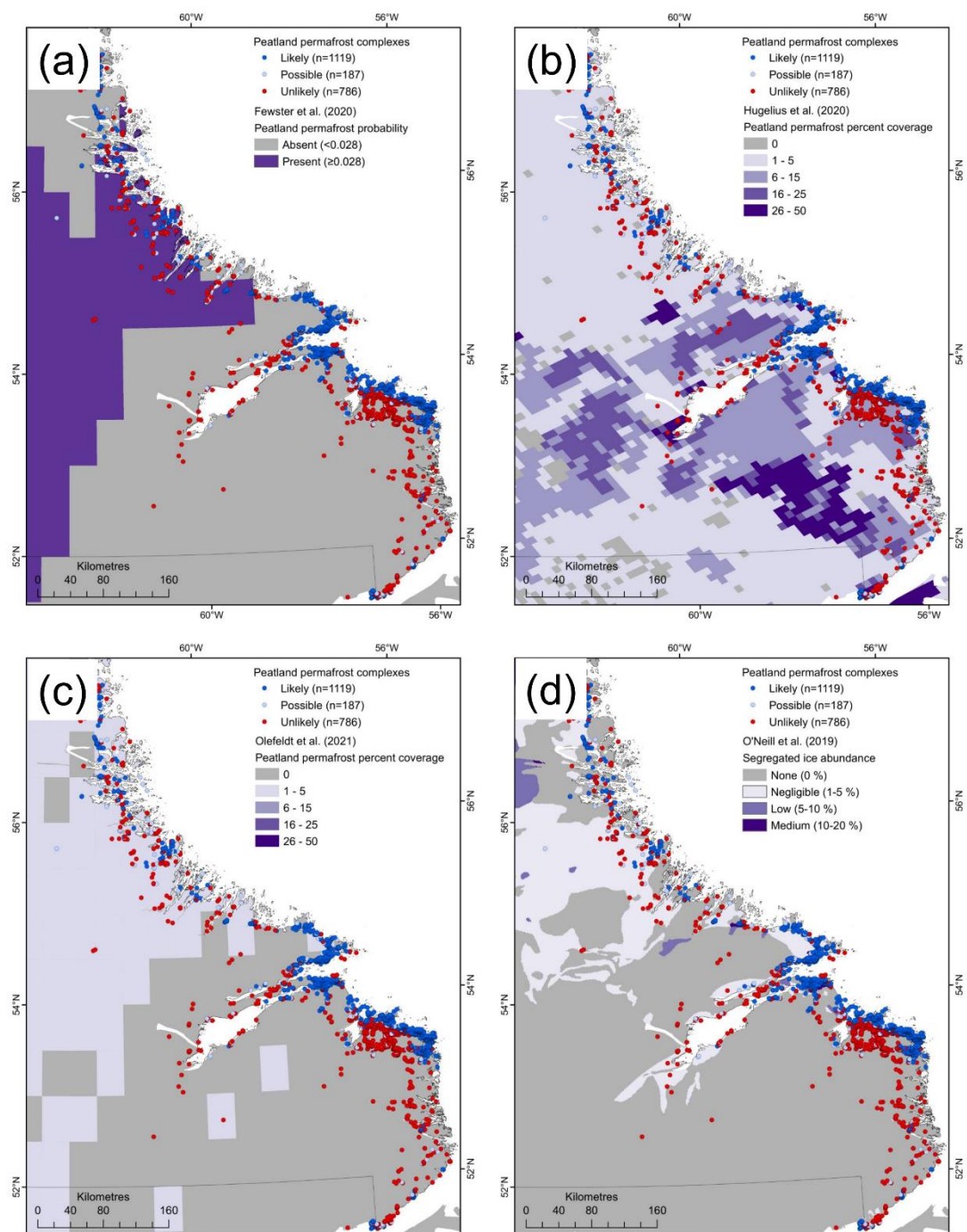

**Figure 7. Comparison of inventoried peatland permafrost complexes with peatland permafrost presence and percent coverage as**
**modelled by (a) Fewster et al. (2020); (b) Hugelius et al. (2020); and (c) Olefeldt et al. (2021) and with segregated ice content as**
**modelled by (d) O'Neill et al. (2019).**

The results of our inventory also suggest that some amendments to existing representations of permafrost distribution
may be required for coastal Labrador. For example, the highest density of peatland permafrost complexes (Figure 4B) was
found near the community of Black Tickle (53.5° N) on the Island of Ponds (2 palsa complexes, 19 mixed palsa and peat
plateau complexes, and 59 peat plateau complexes within 94 km$^2$) (Figure 6), which is currently classified in the isolated
patches of permafrost zone on the Permafrost Map of Canada (Heginbottom et al., 1995) and the no permafrost zone on the
2000-2016 Northern Hemisphere Permafrost Map (Obu et al., 2019) (Supplement Sect. S5). The identification of large swaths
of likely peatland permafrost complexes, including more than 150 peat plateaus, between Cartwright (53.7° N) and Black
Tickle (53.5° N) suggest that the physiography-based Permafrost Map of Canada's limit for the sporadic discontinuous zone
along the Labrador coast (Heginbottom et al., 1997) (Supplement Sect. S5), could reasonably be extended south by ~110 km
from its current position (~53.7° N) (Figure 6B). This southerly extension of the sporadic discontinuous permafrost limit has
previously been suggested by Allard and Seguin (1987) and Payette (2001) who indicated that regional vegetation and
geomorphology favoured permafrost along much of this coastline (Payette, 1983). Unexpectedly, large clusters of likely
peatland permafrost complexes were also identified near the communities of Red Bay (Supplement Sect. S6) and Blanc-Sablon,
both of which are considered to be underlain by little to no permafrost (Heginbottom et al., 1995; Obu et al., 2019) (Supplement
Sect. S5). A 15 km extension of the southern limit of the Permafrost Map of Canada's isolated patches permafrost zone to
include the Blanc-Sablon region would better reflect contemporary permafrost conditions in this area, especially given that
permafrost has been previously detected in mineral soils in the community and in surrounding peatlands below the marine
limit (Dionne, 1984).

**5.3 Challenges and limitations of a point-based inventory of peatland permafrost complexes in coastal Labrador**

The most challenging aspects of the inventorying process involved interpreting peatland permafrost presence in
isolated WOIs containing small landforms, while in the case of more obvious peatland permafrost features, there were at times
difficulties in determining distinct wetland boundaries (Figure 2). However, we believe that these issues were mitigated
through the inclusion of multiple mappers, which facilitated the development of a large initial database and reduced the
potential omission of prospective WOIs. The consensus-based review process that followed was designed to minimize the
inclusion of false positives in the final dataset of 1119 likely peatland permafrost complexes, but we recognize that this
conservative approach may have resulted in the exclusion of some complexes. At the northern end of the study area, where
other types of periglacial landforms become more common, misclassification of palsas for other elevated periglacial landforms
may have contributed to the designation of a higher number of possible peatland permafrost complexes. It is certainly possible
that some segregated ice mounds with less than 40 cm of overlying peat (i.e., lithalsas) may have been included in the inventory,
particularly near the northern end of the study area where wetlands are less abundant and peat deposits may be thinner
(Supplement Sect. S1). This suggests that the definition of peatlands, as wetlands containing at least 40 cm of surface peat
(Tarnocai et al., 2011), and its application to palsas and lithalsas, can introduce some ambiguity during inventorying.

While other inventorying approaches, including grid-based methods (Ramsdale et al., 2017; Gibson et al., 2020, 2021;
Borge et al., 2017), were considered, a point-based inventory was ultimately developed for this study. The implementation of
a grid-based approach with delineation of individual landforms for each WOI could have been useful for estimating ground
ice content, thermokarst potential, carbon content, and overall permafrost coverage, but the purpose of this study was to
generate an initial inventory to guide future research that will facilitate quantitative assessments of peatland permafrost
distribution and coverage in these regions. Our field experience in the region suggests that areal delineations of peatland
permafrost complexes in coastal Labrador will require extensive validation, and it is unlikely that even experienced permafrost
mappers could accurately map the extents of permafrost throughout some complexes without extensive field investigations.
Despite the above limitations, our inventory allowed for the incorporation of dedicated, co-located field- and imagery-based
validation information. Post-validation adjustments to the inventory, including reclassification of 39 WOIs highlights the
importance of ground-truthing in remote sensing- or modelling-based periglacial landform inventories.

Owing to a lack of prior field-based assessments of permafrost conditions in Labrador, it was also difficult to
independently validate our peatland permafrost inventory results. However, a detailed aerial photograph- and field-based
Ecological Land Classification (ELC) survey undertaken in the late 1970s did cover a subset of our study area in southeastern
Labrador (Environment Canada, 1999). The ELC identified a total wetland area of 666 km² which was at least partly covered
by inventoried peatland permafrost landforms (Figure 8). Comparison with the present study showed that mappers identified
peatland permafrost complexes in 23 of the 24 contiguous ELC wetland areas indicated as containing palsas. Examination of
the one remaining ELC peatland permafrost-containing wetland area revealed the presence of irregular ponding patterns
indicative of thermokarst and elevated landforms that could be peatland permafrost but, due to their small size, would require
in situ field visits for validation. Some of the inventoried likely peatland permafrost complexes that were not captured as part
of the peatland permafrost areas from the ELC were instead classified in other wetlands, like string bogs, and in raised marine
terrain units. Overall, the results of our inventory are in good agreement with the limited previous overlapping field
investigations and inventorying efforts from the ELC.

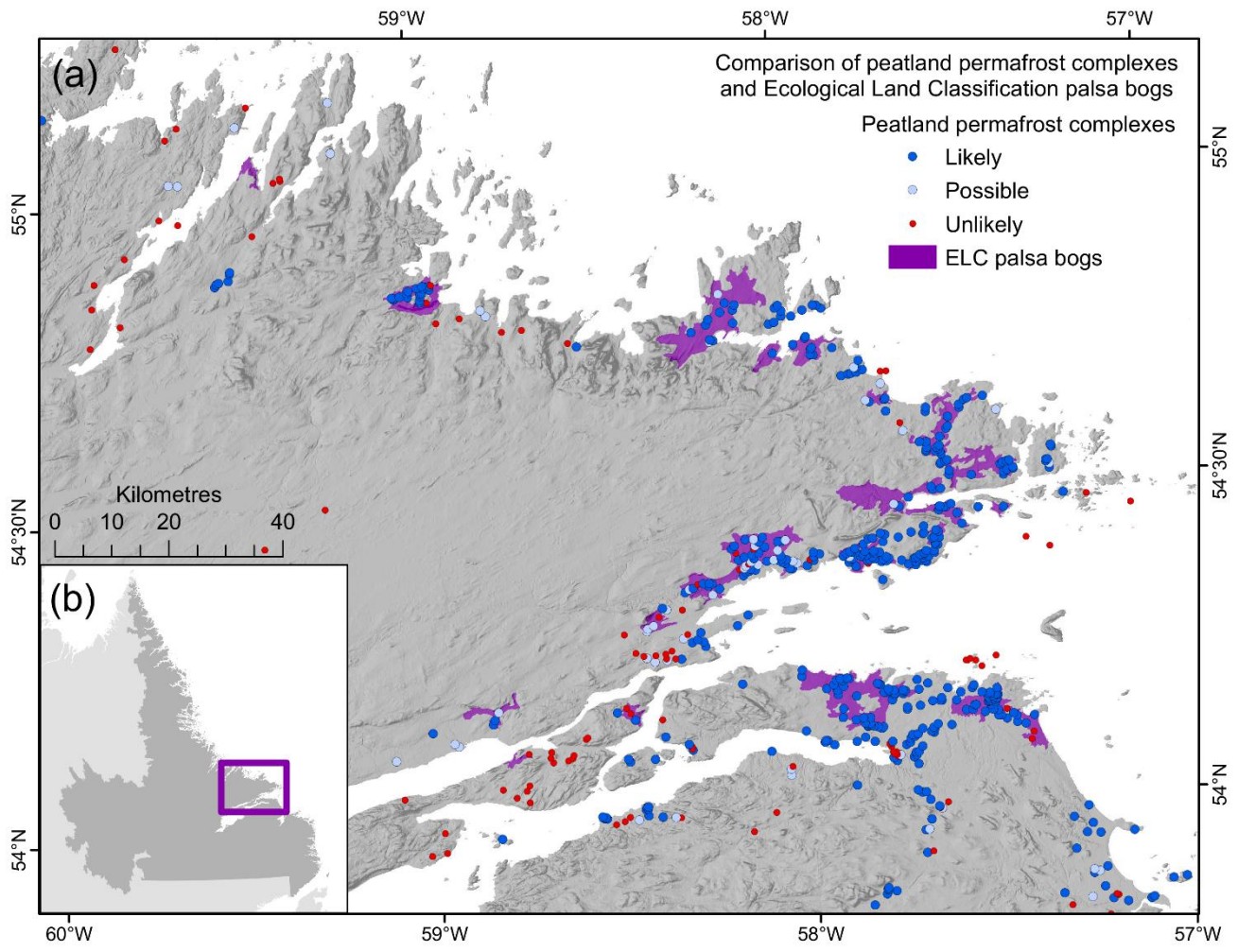

**Figure 8. (a) Comparison of inventoried peatland permafrost complexes with palsa bog regions identified in the Ecological Land**
**Classification (ELC) survey (Environment Canada, 1999); (b) Inset map showing the extent of the peatland permafrost area that**
**was mapped in the ELC.**
**6 Conclusions**
This study provides the first detailed point inventory of peatland permafrost landforms along the Labrador Sea
coastline. Using high-resolution satellite imagery and extensive field- and imagery-based validation efforts, we applied a multi-
stage, consensus-based inventorying approach to identify 1119 likely peatland permafrost complexes. Peatland permafrost
complexes were primarily found in lowlands on outer coasts, spanning from 51.4° N to 58.6° N, with the largest clusters of
complexes occurring ~110 km south of the previously mapped limit of sporadic discontinuous permafrost in northeastern
Canada (Heginbottom et al., 1995).

Comparisons between our point inventory results and existing wetland, peatland, and peatland permafrost distribution products reveal major discrepancies between this study and prior estimates of peatland permafrost in Labrador with implications for ground ice content (O'Neill et al., 2019), thermokarst potential (Olefeldt et al., 2016), and carbon content (Hugelius et al., 2014). Significant advances in the development of relevant datasets on surficial materials, marine limits, peatland distribution, and peat ages and thicknesses, along with field-based advances in climate monitoring for cloud cover, fog, and snow, are critically needed to better characterize northern coastal regions like Labrador. Our results highlight the importance of field-based validation for periglacial landform mapping and modelling and of considering physiography and geomorphology for accurate representations of peatland permafrost in larger scale spatial products. The significant underestimation of peatland permafrost along the Labrador Sea coastline shown in this study should inform future permafrost, peatland permafrost, and carbon content mapping efforts, infrastructure and climate change adaptation strategy development, and wildlife management considerations for Labrador and other northern coastal regions.

**Data availability.** Likely and possible peatland permafrost locations from the coastal Labrador peatland permafrost complex inventory are freely available for download from Nordicana D (Wang et al., 2022).

**Author contribution.** YW and RW designed the study and drafted the manuscript. YW led the raw data collection and the data analysis. RW contributed to raw data collection and data analysis and was the PI for the NSERC Discovery Grant supporting peatland permafrost research activities in Labrador. JB contributed to raw data collection and data analysis. AF and RT contributed to raw data collection. MP coordinated the collection of helicopter video footage. JB, AF, RT, and MP reviewed and contributed edits to the manuscript.

**Competing interests.** The authors declare that they have no conflict of interest.

**Acknowledgements.** The authors would like to acknowledge the Nunatsiavut Government (Rodd Laing), the Nunatsiavut Research Centre (Carla Pamak, Michelle Saunders), the NunatuKavut Community Council (Bryn Wood, George Russell Jr., Charlene Kippenhuck), and the Innu Nation (Jonathan Feldgajer, Jack Penashue) for their guidance on research conducted on traditional Inuit and Innu lands. We thank Caitlin Lapalme, George Way, and Amy Norman of Goose Bay; Freeman Butt and Tanya Barney of Red Bay; Jeffrey and Wendy Keefe of Black Tickle; Barbara Mesher of Cartwright; Jane and Jack Shiwak, Tyler and Harvey Palliser, and Sandi and Karl Michelin of Rigolet; and Caroline Nochasak, Liz Pijogge, Carla Pamak, Kayla Wyatt, Frédéric Dwyer-Samuel, and Patricia Johnson-Castle of Nain for their logistical and in-kind support. We acknowledge Caitlin Lapalme, Victoria Colyn, Kayla Wyatt, Frédéric Dwyer-Samuel, Adrian Earle, Patrick Lauriault, Michaela Smitas-Kraas, and Tara Ryan for their valuable field assistance, and we are grateful to Derrick Pottle, Tyler Palliser, Martin Shiwak, Reginald Maggo, Jeremy Ivany, Martin Andersen, Eldred Andersen, Jeffrey Keefe, Gary Bird, Pat Davis, and Anthony Elson for their boat operation and guiding services. Funding for this research was provided by the Natural Sciences and Engineering

Research Council of Canada, the Northern Scientific Training Program, and Queen's University. We thank Dr. Antoni Lewkowicz and Dr. Luise Hermanutz for many insightful conversations about permafrost and peatlands in Labrador. The manuscript has also benefitted from helpful comments and suggestions from Michelle Saunders, Dr. H. Brendan O'Neill, and Dr. Hanna Lee, and we thank two anonymous referees and Dr. Steve Kokelj for their constructive reviews.

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
