# Peer review of "Significant underestimation of peatland permafrost along the"

_The Cryosphere, 2022_

## Author Comment (AC1)

**AUTHOR RESPONSES TO REFEREE 1 COMMENTARY ON MANUSCRIPT 2022-38**

Manuscript ID#: **2022-38**

Title: **Significant underestimation of peatland permafrost along the Labrador Sea coastline**

First Contact: **Yifeng Wang**

Second Contact: **Robert Way**

**REFEREE 1**

*[Authors' Response]: We thank Referee 1 for taking the time to provide helpful comments on our manuscript. We have responded to each comment below and have made corresponding changes to the revised manuscript.*

**COMMENTS TO THE AUTHORS**

Ln 48-49 – "… suggest that peatland permafrost is more abundant along the coast than in the interior". This seems to be an important point. Does this study confirm this suggestion? Does the study adequately cover the interior peatlands, or focus primarily on the coast? Despite an underestimation of the coastline peatlands, does this study conclude that permafrost is more abundant along the coast than within the interior?

*[Authors' Response]: We have reworded this sentence to clarify that a combination of historical and ongoing use of coastal peatland permafrost environments and a compilation of observations from academic literature of peatland permafrost in coastal Labrador suggest that peatland permafrost is abundant and present along the coast. Prior research in the region, including early works by Roger Brown (1975; 1979), and more recently by Way and Lewkowicz (2016; 2018) and Way et al (2018), suggest a relative absence of peatland permafrost in the interior.*

Ln 65 Another important point. To be clear, the study is a point-based inventory. Does this mean that peatland areas are not outlined, and that no coverage of their extent presently exists? In this case, we do not know the individual area or total area of these peatlands.

*[Authors' Response]: This is correct. Based on our extensive experience from field validation, we are not confident that individual permafrost features can be reliably traced in many regions using the imagery available to us. Peatlands themselves are abundant along the coast but tracing out peatlands is beyond the scope of this study which is focused on peatland permafrost landforms. It should be noted that in Figure S1 (Supplement Sect. S1), we have shown that prior efforts to delineate the distribution of wetlands or peatlands in coastal Labrador show considerable disagreement and thus we do not feel they are reliable enough to use for areal quantification purposes. We have clarified the point-based nature of the inventory in the Abstract, the Introduction, and the Methods. The peatland permafrost complexes were not outlined, so this inventory does not provide any information on the individual area of peatland permafrost complexes or on the total area of peatland permafrost complexes in Labrador. However, we have conducted and included an additional analysis, in which we have classified each likely and possible peatland permafrost complex type as palsa, peat plateau, or mixed (palsa and peat plateau). This information has been included in the main manuscript (Figure 6) and will also help provide us with a solid platform for future area-based analyses, especially given the more*

*extensive permafrost coverage of peat plateaus relative to palsas. We have discussed some of the limitations of the point-based nature of the inventory in Section 5.3 (Challenges and limitations of a point-based inventory of peatland permafrost complexes in coastal Labrador).*

Figure 4d – does the distribution of peatland permafrost landforms by MAAT say anything about past or present conditions in terms of temperature, for their development? That is to say, why does frequency decrease with cooler MAATs? What is the optimal MAAT for their formation?

*[Authors' Response]: This is an excellent question, and we have made some amendments to the text to help clarify. We believe that there are several reasons which explain the discrepancy being noted here. First, peatlands are more extensive in locations with higher MAATs; therefore, the pool of potential locations for peatland permafrost is larger at these MAATs. Second, the importance of the proximity to the Labrador Sea coastline that we have noted for peatland permafrost complexes also shifts these features towards warmer MAATs as compared to if they were found at higher elevations in the interior. We have presented a general description of the distribution of wetlands and peatlands in Section 2.2 (Physical environment) and in the supplement (Supplement Sect. S1).*

Do unlikely peatland permafrost landform areas say anything about past or recent loss of permafrost? Did unlikely areas have permafrost in the past or did they develop without permafrost?

*[Authors' Response]: This is an area of future research that our research laboratory is exploring. We are not confident at this point deciding as to whether these locations previously had permafrost or not, but we do think it is likely that some did, given the presence of small bodies of water that resemble thermokarst ponds in some of these wetlands. Future work using aerial photography from the late 1940s and the early 1990s and satellite imagery from the 2020s will help to answer these important questions. All WOIs were initially identified as peatland complexes that have the potential to contain peatland permafrost and that are worthy of field- or imagery-based validation and additional review.*

Ln 71 Study Area. What is the actual study area – being that area that is encompassed by this study? This section suggests that the study area is all of Labrador – suggesting that the inventory of wetlands of interest using satellite imagery is to cover all of Labrador. If this is not the case, then a specific section that defines the actual study area is necessary. On a map of Labrador, authors should show the actual area covered by their study, otherwise this is rather misleading as it seems that all of Labrador is the study area and has been examined. Suggest the inclusion of a Section 2.4 entitled "limits of study area" which clearly shows and defines the spatial limit that this survey encompasses. At the same time, some statement on what this implies is important as it appears that the study only identifies and attempts to validate peatland permafrost along the coast of Labrador, but not inland.

*[Authors' Response]: We agree with this comment and have modified Figure 1 to include an outline for the study area, corresponding to the area within 100 km of the Labrador Sea coastline. As suggested, we have also included a Section 2.4 entitled "Inventory extent" which describes the extent of the inventory and the justification for our focus on the coast. The few sites included originally outside this domain were largely based on prior field experience in those regions, but their inclusion distracts from the broader intent of this study, so they have been removed.*

Ln 94 Permafrost distribution. This figure should include the outline of the study area within which the surveys were conducted. In this way, readers will be aware of the area in which the study may

attempt to validate permafrost distribution. It seems, in fact, that the results of this study should be sufficient, based on observations, to redefine the distribution of permafrost zones along the coastline based on its findings. This could be an added objective and it seems reasonable that if the surveys found permafrost peatlands along the coastline but not inland – that the extent of sporadic permafrost could be extended along the coastline and shown as an additional result in this study. If the authors feel they do not have enough evidence in their study to extend the sporadic zone at present, then they should suggest what else is needed to do so either in the discussion or the conclusion.

*[Authors' Response]: As suggested, we have modified Figure 1 to include an outline of the study area, corresponding to the area within 100 km of the Labrador Sea coastline.*

*We have conducted and included an additional analysis, in which we have classified each likely and possible peatland permafrost complex type as palsa, peat plateau, or mixed (palsa and peat plateau). This information has been included in the main manuscript (Figure 6) and helps to inform our discussions on amendments to the southern limit of the sporadic discontinuous permafrost zone in Labrador (Section 5.2, Implications for peatland permafrost and permafrost distribution in northeastern Canada). We feel that a high density of permafrost observations in locations that are currently classified as "isolated patches" or "no permafrost" does provide a compelling argument for this amendment, particularly if they are observations of peat plateaus, which correspond to large areas of permafrost. As suggested, we have included the revision of the current limits of permafrost distribution zones in coastal Labrador as an objective in the Introduction.*

Ln 115 Methods. Again, it is important to define the area along the Labrador Sea and Gulf of St. Lawrence coastline that is actually covered by this study. In essence, the study only identifies and attempts to validate peatland permafrost within these areas – not within all of Labrador. Figure 1 can be used to show contiguous survey areas along coast and can also indicate that inland point features outside of these areas were also investigated.

*[Authors' Response]: We have modified Figure 1 to include an outline of the main study area, corresponding to the area within 100 km of the Labrador Sea coastline. Strictly speaking, the lack of features in the interior is due to the lack of features, not the study design. There are members of our team (e.g., R.G. Way) who have spent decades on the land in the interior of Labrador without encountering these features regularly. Nevertheless, for the purposes of streamlining and outlining a clearer study design, we have made the requested changes.*

Ln 115 Methods. The methods section needs to discuss issues of scale. Specifically, how large / how small an area was identified on satellite imagery. Not only the resolution of the imagery, but what is the minimum size of a permafrost peatland that was counted as a peatland complex and, similarly, how large. It seems that this study did not outline peatland permafrost complexes, but simply identified them as point-based features. Does this mean that each feature was contiguous, or does this include multiple features close together. Similarly, how far away does another feature need to be to be counted as a separate feature? As these are indicated only as point features, it is important to provide some methodological constraints on how a feature was included (minimum size) and how it was differentiated from a separate feature (minimum separation distance). It would be very useful it there were also some insight into the size range of these features – even if they were mapped only as point features.

*[Authors' Response]: Information on the resolution of the imagery is included in both Section 3.1 (Data sources) and Section 3.2.1 (Identifying wetlands of interest (WOIs)), and we have provided additional information in Section 3.2.1 (Identifying wetlands of interest (WOIs)) on the range in size of the WOIs. We appreciate the suggestion to include information on the minimum separation distance between WOIs, but we run into challenges with this in relation to differences in the physiographic and geomorphologic characteristics of wetlands across Labrador. Based on field investigations, we have noticed that wetlands are more widespread in southern Labrador, but they tend to be smaller (as small as ~0.2 km2). Near the northern end of the study area, wetlands are less common, but the ones that are present are often very large (as large as ~3.5 km2). The largely geomorphological approach that mappers applied during the identification and mapping stage, based on differences in drainage, vegetation, and morphology between prospective wetland complexes, make it difficult to report a standard minimum separation distance between all WOIs within this study. We have also generated a new figure that shows how the delineation of the WOIs was interpreted and have included this figure as Figure 2 in the main manuscript.*

It is not generally clear why a point-based inventory was approached, rather than outlining the potential peatland permafrost terrain units. Perhaps, at least, it could be stated why point-based mapping was undertaking rather than defining polygons and areas.

*[Authors' Response]: Our focus was on identifying peatland permafrost landforms in wetlands and not on characterizing wetlands with peatland permafrost. While the distinction may not be obvious, we believe that our approach is all that can currently be produced given the uncertainties in mapping peatland permafrost distribution in our region. We also believe that the current lack of knowledge on these features in the region necessitates building a step-by-step baseline understanding. We have provided additional justification for the point-based nature of the inventory and have also included a discussion of some of the limitations of this approach compared to grid-based or areal-based inventorying in Section 3.2.1 (Identifying wetlands of interest (WOIs)) and Section 5.3 (Challenges and limitations of a point-based inventory of peatland permafrost complexes in coastal Labrador). It should be noted that unlike many regions elsewhere in Canada, there are no prior observations of permafrost in most of the regions we are describing.*

As a note, it would have been beneficial for the authors to have perhaps differentiated the sizes of the peatland permafrost terrain into a least "small", "medium" and "large" peatland units with some type of catagorization. For example, in Figure S9 it becomes clear that permafrost peatlands are of different sizes, and may benefit from differentiation. In Figure S10 it is not really clear how one peatland unit is differentiate from another as they are shown only as point features and the boundaries of each a not easily to distinguish. Again, a simple differentiation of the size of each in catagorization would have been beneficial.

*[Authors' Response]: We have provided additional information in Section 3.2.1 (Identifying wetlands of interest (WOIs)) on the range in size of the WOIs. We have also generated a new figure that shows how the delineation of the WOIs was interpreted and have included this as Figure 2 in the main manuscript. While the peatland size is relevant to a wetland mapping initiative, we are worried about an implication that larger peatlands may have more permafrost area when that is not something that we can evaluate at this point. Instead, we have conducted and included an additional analysis, in which we have classified each likely and possible peatland permafrost complex type as palsa, peat plateau, or mixed (palsa and peat plateau). This information has been included in the main manuscript (Figure 6) and will be useful for future area-based analyses given the more extensive permafrost coverage by peat plateaus relative to palsas.*

Ln 208-212 Even though areas were identified only as point features, something about their size should be included. What was minimum size, what was maximum size? Even point features have separation distances, so what was the minimum separation distance between features?

*[Authors' Response]: We have provided additional information in Section 3.2.1 (Identifying wetlands of interest (WOIs)) on the range in size of the WOIs. We appreciate the suggestion to include information on the minimum separation distance between WOIs, but we run into challenges with this in relation to differences in the physiographic and geomorphologic characteristics of wetlands across Labrador. Based on field investigations, and informed by many wetland and peatland distribution maps presented in the supplement (Supplement Sect. S1), we have found that wetlands are more widespread in southern Labrador, but they tend to be smaller (as small as ~0.2 km2). Near the northern end of the study area, wetlands are less common, but the ones that are present are often very large (up to ~3.5 km2). The largely geomorphological approach that mappers applied during the identification and mapping stage and these differences in wetland characteristics make it difficult to report a standard minimum separation distance between all WOIs within this study.*

Ln 280-285. Discussion regarding distribution of permafrost peatland complexes is intriguing, and also opens up additional discussion. Where are data showing which peatland complexes lie below marine limit, and which are above? This is alluded to but not shown.

*[Authors' Response]: We agree with the reviewer that providing information about the distribution of peatland permafrost complexes relative to the marine limit would be extremely relevant and useful for this study and for understanding overall permafrost distribution in Labrador. Unfortunately, there is no existing marine limit or marine sediment dataset for the entire coast of Labrador, so it is difficult to provide an estimate of the elevation of each of the peatland permafrost complexes relative to the local marine limit. This is a typical issue in Labrador where there is a paucity of baseline information compared to other regions (e.g., the Northwest Territories). We have estimated the local marine limits for as much of our study area as possible using inverse distance weighted interpolation from a series of observations that were compiled in Dyke et al. (2005) and have presented this information in the supplemental (Supplement Sect. S3), but we have not presented it as part of the main manuscript given that the interpolation does not cover our entire study area. We have also included additional information in Section 5.1 (Distribution of peatland permafrost in Labrador) describing the lack of available data on marine limits or marine sediments.*

The issue of deglacial history and marine recession history are relevant here, in terms of defining the oldest terrestrial age surface in the study area and, thus, oldest peatlands. It appears that deglaciation of the region was from as early as 11 ka BP, along the coastline and then younger moving inland to about 7 ka BP. At the same time, marine recession was occurring in the southern areas along the coastline. Presumably, along the coastline at certain elevations deglaciation and marine recession were the earliest, and these are the oldest peatlands. So – are the oldest peatlands generally also the ones with likely permafrost? Are they thickest, do they have the most syngenetic ground ice? It would be useful to tie the history of marine recession and deglaciation into this discussion a bot more. At present, this is portion of the discussion very limited and is worthy of further consideration.

*[Authors' Response]: We agree with the reviewer on the relevance of deglacial history, marine recession, and peatland age to peatland permafrost distribution, but we are unfortunately limited*

*by the lack of available information on these variables. We have included additional information in Section 5.1 (Distribution of peatland permafrost in Labrador) describing the importance of considering peatland initiation timing and peat deposition rates for peatland permafrost distribution, based on their impacts on peat thickness and the thermal offset.*

Ln 287-299. Again, there seems to be more to say here when speculating on the history of peatland intiation ages within the study area – which most of these products/datasets do not take into consideration (and presently, the authors do not either). Admittedly, few peatland initiation ages exist in the region, though theoretically the youngest may be constrained to near the coast. The authors might consider referring to the following articles as a starting points on understanding peatland ages in the region and their possible influence on permafrost peatland distribution:

Gorham, E., Lehman, C., Dyke, A., Janssens, J. and Dyke, L., 2007. Temporal and spatial aspects of peatland initiation following deglaciation in North America. Quaternary Science Reviews, 26(3-4), pp.300-311.

And:

Dyke, A.S., Giroux, D. and Robertson, L., 2004. Paleovegetation Maps of Northern North America, 18 000 to 1 000 BP. Geological Survey of Canada.

*[Authors' Response]: We appreciate this comment and have included additional information in Section 5.1 (Distribution of peatland permafrost in Labrador) mentioning the importance of considering peatland initiation timing, peat deposition rates, and peat thickness for peatland permafrost distribution.*

Ln312-313: It seems that this study could go a step further by outlining the proposed extension of sporadic permafrost based on their results. Providing an additional Figure 7 with proposed areas of sporadic permafrost would be a useful addition and seems reasonable based on the extent of the study and the results.

*[Authors' Response]: We appreciate this suggestion. We do recognize that one's interpretation of the isolated patches of permafrost zone, as a zone within which less than 10% of the area is underlain by permafrost, can complicate areal estimates of permafrost coverage given this distribution zone's lack of a lower threshold. Despite these challenges, we do believe that a high density of permafrost observations in locations that are currently classified as "isolated patches" or "no permafrost" provides a compelling argument for the southerly extension of the sporadic discontinuous permafrost zone. To support this argument, we have conducted and included an additional analysis, in which we have classified each likely and possible peatland permafrost complex type as palsa, peat plateau, or mixed (palsa and peat plateau). This information has been included in the main manuscript (Figure 6) and is very useful for informing discussions regarding the potential southward extension of the sporadic discontinuous permafrost zone (Section 5.2, Implications for peatland permafrost and permafrost distribution in northeastern Canada). For example, we have used this information to help inform the proposed location for a new southern limit of the sporadic discontinuous permafrost zone in Labrador, and as suggested, we have included this in the main manuscript as Figure 6B.*

Ln330-333: This may warrant an additional sentence or two for clarification. What is the basis for mis-identification based on? For example, most maps in Fig S5 show greater abundance of wetland or peatland areas in the south than in the north. Is it the absence of mapped peatlands along the coastline in these inventories that leads author's to suggest that their identified areas here may not

be peatland permafrost, but instead lithalsa's? Or did field visits (Fig. 2) along the northern coastline confirm that these were lithalsa's or in fact peatland permafrost? In general, the absence of peatlands shown in Fig. S5 suggests that either there are few peatlands here, or they are too small to be mapped at that scale.

*[Authors' Response]: We agree and have included additional sentences in Section 5.3 (Challenges and limitations of a point-based inventory of peatland permafrost complexes in coastal Labrador) to clarify that the potential inclusion of lithalsas in the inventory is linked to the requirement for peatlands (and therefore peatland permafrost landforms) to contain 40 cm of peat. Segregated ice mounds found in wetlands with less than 40 cm of overlying peat may have been included in the inventory, particularly in the northern end of the study area where wetlands are less abundant and peat deposits are thinner.*

Figure 6. Reference source for this map seems odd "audio tape?". Whereas it is interesting to show palsa bogs mapped by ELC here, were there other terrain types related to peatlands that were mapped too? There seems to be a good agreement between the mapped palsa bogs and peatland permafrost, but what were other areas mapped as? Were these peatland areas that did not contain permafrost or other terrain types? Could be discussed in text if not in figure itself.

*[Authors' Response]: We agree that it is an odd reference, but it seems to be the only available resource that describes the survey. The audio tape transcript is available for download from Natural Resources Canada, and we have included the download link for the transcript document to the reference. As suggested, we have included a sentence in the text to describe some of the other terrain units from the ELC within which the remaining likely peatland permafrost complexes were found.*

This study seems almost purposefully vague about existing weather and climatic conditions occurring within the areas of identified permafrost peatland terrain. Given the adherence of these areas to the Labrador coastline, it is indeed interesting to speculate to what extent a maritime climate influences the distribution of permafrost across the study area. The authors allude to conditions of fog, cloud cover, snowpack and wind being potential factors in their distribution. Presumably, these factors are being examined in site-specific studies. The authors could elaborate somewhat further, in the discussion, and most certainly in the conclusion, for the need to investigate local climatic conditions that may support the presence of permafrost in these areas. In a way, this is similar to the examination of the role of inversions in some mountainous environments for sustaining permafrost. It would be suitable for the authors to provide some insight into the intent and value of local studies to understand the distribution of contemporary permafrost further. In addition, such work could aid in more accurately determining extent of sporadic permafrost along this maritime area.

*[Authors' Response]: As suggested, we have included sentences in both the Discussion (Section 5.1; Distribution of peatland permafrost in Labrador) and the Conclusion describing the need for additional local, field-based investigations into the role of certain climatic variables on peatland permafrost distribution and persistence.*

Figure S3. Not sure that depicting only locations of non-peatland permafrost locations is useful. Perhaps better to include both those that did as well as those that did not.

*[Authors' Response]: Following previous suggestions, we have defined our study area more clearly as the area of Labrador and adjacent parts of Quebec that fall within 100 km of the*

*Labrador Sea coastline. We have described in Section 2.4 (Inventory extent) our justification for our focus on the coast, so we have removed this figure from the supplemental material as it is no longer necessary.*

Suggest adding "northeastern Canada" to the end of the title

*[Authors' Response]: We have added "in northern Canada" to the end of the title.*

Ln 12 Change "maps" to "depictions"

*[Authors' Response]: We have changed "maps" to "estimates".*

Ln 21 Ditto

*[Authors' Response]: We have changed "maps" to "estimates".*

Ln 27 consider replacing "perennially frozen ground" with "permafrost"

*[Authors' Response]: We appreciate the suggestion, but we have kept the text as is to avoid using the word "permafrost" too many times in the same sentence.*

Ln 41 delete "they"

*[Authors' Response]: We agree and have deleted "they".*

Ln 43 consider replacing "have suggested that peatland permafrost is present" with "have depicted peatland permafrost as present"

*[Authors' Response]: We agree and have changed "have suggested that peatland permafrost is present" to "have depicted peatland permafrost as present".*

Ln 46 change "is" to "are"

*[Authors' Response]: We appreciate the suggestion, but we have kept the text as is.*

Ln 58 change "have  been" to "are"

*[Authors' Response]: We have changed this sentence to "Previous peatland permafrost mapping in Labrador has…".*

Ln 60 change "and no" to "with no"

*[Authors' Response]: We agree and have changed "and no" to "with no".*

Ln 60 change "efforts have been completed"  to "effort completed"

*[Authors' Response]: We agree and have changed "efforts have been completed" to "efforts completed".*

Ln 75 provide location of coldest MAAT (-11.9C) and  warmest MAAT (+1.5C) for context and, if possible, so locations on Figure 1.

*[Authors' Response]: We have included details describing the locations of the lowest MAAT, in the Torngat Mountains, and highest MAAT, near the community of Blanc-Sablon.*

Ln 73-78. Unless provided elsewhere, indicate proportion of snowfall versus rainfall and range in total precipitation.

*[Authors' Response]: We have included a sentence describing the maximum precipitation, based on the CHELSA dataset, and the annual proportion of snow to rain at two Environment Canada weather stations at opposite ends of our study area over the 1981-2010 climate normal.*

Ln 87 How can glacial till be deposited following retreat of the Laurentide Ice Sheet, except by another glacial/glaciation? Explain, rephrase or delete.

*[Authors' Response]: We agree and have removed "and following" from the sentence.*

Ln 96 Try to keep spelling of words like "archaeological" and "paleogeographic" consistent. Decide on preferred spelling and use it throughout.

*[Authors' Response]: We agree and have changed "archaeological" to "archeological".*

Ln 141 change "that exceeded" to "exceeding"

*[Authors' Response]: We agree and have changed "that exceeded" to "exceeding".*

Ln 177-178. Change "wetland complex by wetland complex" to "WOI" if appropriate.

*[Authors' Response]: We agree and have changed "wetland complex by complex" to "WOI by WOI".*

Ln 188 Change "was" to "were".

*[Authors' Response]: We agree and have changed "was" to "were".*

Ln 189 Delete "of WOIs"

*[Authors' Response]: We have deleted this sentence as suggested by Referee #3.*

Ln 191 Delete "that was"

*[Authors' Response]: We agree and have removed "that was".*

Ln 251 95 % - remove space.

*[Authors' Response]: We appreciate the suggestion but have retained the space to ensure that all units in the text are preceded by a space.*

Ln 262 Delete "In this, study, we demonstrated that". Start sentence with "Peatland permafrost …". Reference Figure 4b at end of sentence.

*[Authors' Response]: We agree and have started the sentence with "Peatland permafrost ...". We have referenced Figure 5B at the end of the sentence.*

Ln 265 Provide reference to a figure as supporting evidence.

*[Authors' Response]: We agree and have provided reference to Figure 4 as supporting evidence.*

Ln 535 Reference seems incomplete. Nordicana D?

*[Authors' Response]: Thank you, we have reformatted the reference.*

Ln 538-539 Reference incomplete.

*[Authors' Response]: Thank you, we have reformatted the reference.*

---

## Author Comment (AC2)

**AUTHOR RESPONSES TO REFEREE 2 COMMENTARY ON MANUSCRIPT 2022-38**

Manuscript ID#: **2022-38**

Title: **Significant underestimation of peatland permafrost along the Labrador Sea coastline**

First Contact: **Yifeng Wang**

Second Contact: **Robert Way**

**REFEREE 2**

*[Authors' Response]: We thank Referee 2 for taking the time to provide helpful comments on our manuscript and supplemental materials. We have responded to each comment below and have made corresponding changes to the revised manuscript and supplemental materials.*

**COMMENTS TO THE AUTHORS**

Some consideration of the scale of existing maps of permafrost and peatland distribution compared to the scale of the authors' study is required in the analysis and formulation of the main conclusions regarding adjustment of existing maps for southern Labrador. The maps used for comparison are at a smaller scale (national and circumpolar) than the more local to sub-regional scale mapping presented in the MS. Many of the maps used (or the ones used to develop them) will have minimal mapping units so that the characteristics of smaller units will not be shown on the map. This would be the case for example, with the Heginbottom (1995) which is at a scale of 1:7 million, and to some extent O'Neill et al. (2019) which utilizes similar scale maps in its development. It is therefore not surprising that your results would be a bit different. At a national or circumpolar scale, the 15 km that the authors' suggest the southern permafrost boundary should be extended, is within the precision of these maps. One of the points that could be made is that the application of national and circumpolar scale maps is not really appropriate for addressing sub-regional to local scale issues including those related to plant and animal habitat or infrastructure scale integrity as has been done in a number of other studies. Although the authors do seem to hint at issues of scale, this aspect could be strengthened in the paper.

*[Authors' Response]: We agree with the reviewer that the application of national and circumpolar scale maps is not appropriate when addressing sub-regional to local scale issues, but these smaller-scale maps are unfortunately often still used to inform local infrastructure and land use initiatives, or at least to characterize and provide context to a study area, especially in the absence of more appropriate or relevant datasets. However, it is arguably more important to recognize that the Permafrost Map of Canada and the International Permafrost Association's Circumarctic Map of Permafrost and Ground Ice Conditions was not derived from an actual areal calculation of permafrost but is rather based on a holistic assessment of existing permafrost information in conjunction with physiographic data (Heginbottom et al., 1997; Heginbottom, 2002; Zhang et al., 2008). We believe that the primary reason that the sporadic discontinuous zone was not extended farther south along the coast of Labrador was because the very few studies from Labrador that might have informed the development of this map (the Ecological Land Classification (Environment Canada, 1999) and Roger Brown's investigations (1975; 1979)) did not cover the areas that we are currently describing as having a high density of peatland permafrost complexes. Issues of scale certainly do apply to maps derived using explicit areal calculations and modelling,*

*but for a holistic mapping effort like the IPA map, we believe the comparison is fair. We have elaborated on these issues in the supplemental material (Supplement Sect. S5).*

The inventory would appear to consist of point observation of frozen peatlands. It is not clear if the area of these features has also been determined. This would be useful for the comparison to existing permafrost and peatland maps which show distribution in terms of areal coverage rather than location of specific occurrence of features. Although the density of peatland complexes likely containing permafrost (number) per 400 $km^2$ is shown in figure 3b this is not the same as % areal coverage as shown on other existing maps. This makes it difficult to determine whether the results indicate greater occurrence of frozen peatlands than the maps that are used for comparison in the MS (i.e. comparing apples to oranges). Many of the likely or possible occurrences of peatland permafrost complexes are for example within the sporadic or isolated patches zones shown on the Heginbottom et al. (1995) map which means permafrost is more likely than not to be absent and limited to organic terrain in the case of isolated patches. It is difficult to determine from the results presented whether the map presented in the MS indicates a permafrost distribution that is different from the Heginbottom et al. map. Some further discussion is probably required regarding area of the features identified in the inventory.

*[Authors' Response]: We appreciate this comment and have clarified in the Abstract, the Introduction, and the Methods that the inventory is a point inventory. We agree with the reviewer that the occurrence of likely or possible peatland permafrost complexes within the sporadic or isolated patches of permafrost zones suggest that permafrost is likely limited to organic terrain. However, we note that one's interpretation of the isolated patches of permafrost zone, as a zone within which less than 10% of the area is underlain by permafrost, can complicate areal estimates and one's general understanding of permafrost coverage given this distribution zone's lack of a lower threshold. Rather than provide an estimate of the areal coverage of peatland permafrost landforms within each WOI, we have conducted and included an additional analysis, in which we have classified each likely and possible peatland permafrost complex type as palsa, peat plateau, or mixed (palsa and peat plateau). This information has been included in the main manuscript (Figure 6) and will provide us with a solid platform for future area-based analyses, especially given the more extensive permafrost coverage by peat plateaus compared to palsas. We have also included a description of the size range for the WOIs in Section 3.2.1 (Identifying wetlands of interest (WOIs), and we have addressed some of the limitations of the point-based nature of the inventory in Section 5.3 (Challenges and limitations of a point-based inventory of peatland permafrost complexes in coastal Labrador), compared to a grid-based or similar area-based inventory.*

I am somewhat curious as to how the maps for comparison in the main paper (figure 5) were chosen especially the circumpolar maps (Hugleius et al. 2020; Olefeldt et al. 2021) rather than some of those included in the supplementary information. Would the larger scale map of Tarnocai et al. (2011) for example (which I believe also includes information on whether peatlands are frozen), be more suitable for comparison in the main paper.

*[Authors' Response]: The three peatland permafrost distribution products that were included in the main paper are the most recently published estimates for peatland permafrost distribution. Unfortunately, the Tarnocai et al. (2011) dataset suggests that there are no perennially frozen peatlands in any part of Labrador. We have presented a comparison between the inventoried peatland permafrost complexes and the Tarnocai et al. (2011) product in the supplemental material (Supplement Sect. S4).*

Some clarification on the study area is required. It would seem that the focus is on Labrador (coastal Labrador?) but the authors should clarify if the imagery analysis was done for all of Labrador or only specific areas. Also there appear to be observations outside of Labrador and it is unclear which areas outside of Labrador were included in the imagery analysis. A map clearly showing the area for which imagery analysis was done would therefore be useful. For field-based observations, some information on how sites were chosen beyond accessibility is probably required for the reader to understand whether there is any bias in the site selection and validation.

*[Authors' Response]: Prior research in the region, including early works by Roger Brown (1975; 1979), and more recently by Way and Lewkowicz (2016; 2018) and Way et al. (2018), suggest a relative absence of peatland permafrost in the interior. For the purposes of streamlining and making a clearer study design, we have modified Figure 1 to include an outline for the main study area, corresponding to the area within 100 km of the Labrador Sea coastline. We have also included a Section 2.4 entitled "Inventory extent" which describes the extent of the inventory and the justification for our focus on the coast. We have also provided additional information in Section 3.3 (Validation of subset of WOI database) on the access to WOIs for field validation.*

L2 – Title – would it be better to refer to the "Labrador coast"?

*[Authors' Response]: We have added "in northern Canada" to the end of the title, based on a suggestion by Referee #1.*

L30-31 – insert "in temperature" between "offset" and "between" (i.e. be clear that the offset is referring to a difference in temperature). You could also add that it is the difference between the frozen and unfrozen thermal properties that is an important factor.

*[Authors' Response]: We agree and have clarified that it is "a large temperature offset".*

L34 – "assessment of thermokarst…." Is probably better and more inclusive.

*[Authors' Response]: We agree and have changed "predicting thermokarst potential" to "assessing thermokarst potential".*

L50 – O'Neill et al is a national scale map and is based on integration of a national scale surficial map which will not show local scale distribution of peatlands or other organic terrain.

*[Authors' Response]: We have clarified that the ongoing underestimation of peatland permafrost in the region can influence ground ice estimates. We recognize that the O'Neill et al. (2019) product integrates information on surficial materials, paleovegetation, deglaciation, and contemporary permafrost distribution, but we think that ground ice content, thermokarst potential, and carbon content are important to mention in relation to the distribution and sensitivity of peatland permafrost.*

L60 – There is the peatland map and database which I believe is at least partly based on air photo interpretation of Tarnocai at al. (cited in Supplemental Information).

*[Authors' Response]: Unfortunately, the Tarnocai et al. (2011) dataset suggests that there are no perennially frozen peatlands in any part of Labrador. We have presented a comparison between the inventoried peatland permafrost complexes and the Tarnocai et al. (2011) product in the supplemental material (Supplement Sect. S4).*

L275-277 – Way and Lewkowicz (2018) includes ground temperature measurements in Labrador and the thermal offsets for various terrain types. Could you be more quantitative and use these

results to strengthen the point you are trying to make regarding importance of thermal offset. James et al. 2013 ERL also discusses the importance of thermal offset in persistence of permafrost in organic terrain.

*[Authors' Response]: We appreciate the suggestion and have included the approximate thermal offset at peatland permafrost locations from Way and Lewkowicz (2018) to this section of the Discussion to strengthen our argument against the utility of MAAT-based thresholds for predicting peatland permafrost distribution.*

L278-285 (also figure 4) – With respect to associations with elevation, it might be more important to consider whether the area is above or below the marine limit rather than the elevation itself. Given the marine limit varies with latitude, as described in section 2.2, it would make sense to consider the location with respect to the marine limit. For sites below the marine limit, wouldn't the time since emergence be a factor as it would influence age of peatland and also length of time over which ground freezing occurs.

*[Authors' Response]: We agree with the reviewer that providing information about the distribution of peatland permafrost complexes relative to the marine limit would be extremely relevant and useful for this study and for understanding overall permafrost distribution in Labrador. Unfortunately, there is no existing marine limit or marine sediment dataset for the entire coast of Labrador, so it is difficult to provide an estimate of the elevation of each of the peatland permafrost complexes relative to the local marine limit. We have estimated the local marine limits for as much of our study area as possible using inverse distance weighted interpolation from a series of observations that were compiled in Dyke et al. (2005) and have presented this information in the supplemental (Supplement Sect. S3), but we have not presented it as part of the main manuscript given that the interpolation does not cover our entire study area. We have also included additional information in Section 5.2 (Implications for peatland permafrost and permafrost distribution in northeastern Canada) describing the lack of available data on marine limits or marine sediments, and we have included information in Section 5.1 (Distribution of peatland permafrost in Labrador) on the potential role of peat age and peat thickness in peatland permafrost development and persistence through the thermal offset.*

L287-299 – Reference is made to model predictions. It might be better to refer to simulations which would be more inclusive as the various studies mentioned use various approaches including compilation/synthesis of existing information.

*[Authors' Response]: We agree and have changed "models" to "simulations".*

L298 – The surficial deposits are a key factor influencing drainage and accumulation of organic matter as well as formation of segregated ice. You might consider association of peatland permafrost with surficial deposits as has been done for other parameters in figure 4.

*[Authors' Response]: We agree with the reviewer that surficial deposits are very important for understanding peatland permafrost distribution. Surficial materials information for the entirety of Labrador is currently only available at the 1:1,000,000 scale, with some information at the 1:50,000 scale in scattered locations. Our ability to make these kinds of comparisons is unfortunately limited by the availability of surficial materials products at an appropriate scale and will not be possible until significant advances are made in this area by partner institutions or governments. We have included additional information in Section 5.1 (Distribution of peatland*

*permafrost in Labrador) describing the lack of available surficial materials data at a suitable scale for all of Labrador.*

L309-310 - Obu et al. (2019) map represents equilibrium conditions so it doesn't adequately consider past climate history which is important as you have mentioned in the discussion. Permafrost occurrence will be underestimated, especially in the southern portion of the permafrost zone.

*[Authors' Response]: We do not believe that the disagreement with Obu et al. (2019) in our region is due to equilibrium modelling but rather reflects performance issues with their implementation of the TTOP model. Unpublished work by Way and Lewkowicz presented at the Eastern Snow Conference in 2017 showed that discrepancies between TTOP spatial models (e.g., Way and Lewkowicz, 2016) and observations of peatland permafrost along the southern coast of Labrador could largely be reconciled with an improved snow redistribution algorithm and more precise land cover maps. While equilibrium modelling could potentially explain a lack of peatland permafrost in some areas, it is not the primary source of disagreement. We have elaborated on potential issues in the interpretation of TTOP model results in the supplemental material (Supplement Sect. S5).*

L319 – You need to consider the scale of the maps to which you are comparing your results. Heginbottom et al. is a national scale map and is much at a much smaller scale than your study – 15 km on the national scale mapping is likely within the precision of the map.

*[Authors' Response]: We have briefly mentioned the differences in scale between our inventory and the products used for comparison in Section 5.2 (Implications for peatland permafrost and permafrost distribution in northeastern Canada) and have elaborated on issues of scale in the supplemental material (Supplement Sect. S5). However, we also note that the Permafrost Map of Canada and the International Permafrost Association's Circumarctic Map of Permafrost and Ground Ice Conditions were derived from a holistic assessment of existing permafrost information in conjunction with physiographic data (Heginbottom et al., 1997; Heginbottom, 2002; Zhang et al., 2008). We believe that the primary reason that the sporadic discontinuous zone was not previously extended farther south along the coast of Labrador was because the very few studies from Labrador that might have informed the development of this map (the Ecological Land Classification (Environment Canada, 1999) and Roger Brown's investigations (1975; 1979)) did not cover the areas that we are currently describing as having a high density of peatland permafrost complexes. Issues of scale certainly do apply to maps derived using explicit areal calculations and modelling, but for a holistic mapping effort like the IPA map, we believe the comparison is fair.*

L400-401 – Is this a conference presentation with abstract? Provide the conference details and abstract if that is the case

*[Authors' Response]: Thank you, we have reformatted the reference.*

L404-405 – Incomplete citation. Is this an unpublished report?

*[Authors' Response]: Thank you, we have reformatted the reference.*

L406 – Unpublished report, conference presentation? Provide details.

*[Authors' Response]: Thank you, we have reformatted the reference.*

L413-414 – This is NRC Internal Report No. 82 with 1956 publication date.

*[Authors' Response]: Thank you, we have reformatted the reference.*

L415-416 – Incomplete. This is NRC Technical Paper 449

*[Authors' Response]: Thank you, we have reformatted the reference.*

L432 – Is this correct. Seems like an odd reference for a land survey

*[Authors' Response]: We agree that it is an odd reference, but it seems to be the only available resource describing the survey. The audio tape transcript is available for download from Natural Resources Canada. We have included the download link for the transcript document to the reference.*

L434 – van Everdingen is the editor. Also, you should indicate this is an International Permafrost Association publication of the Terminology Working Group

*[Authors' Response]: Thank you, we have reformatted the reference.*

L441 – Is this from the Quaternary Geology of Canada and Greenland. Add missing citation info.

*[Authors' Response]: Thank you, we have reformatted the reference.*

L442 – This is Map 1880A and it should have a doi number (check GEOSCAN https://geoscan.nrcan.gc.ca/ )

*[Authors' Response]: Thank you, we have reformatted the reference.*

L453 Missing information. This is from the National Atlas (5th Edition) Geomatics Canada series number MCR 4177. It also has a doi number (check GEOSCAN https://geoscan.nrcan.gc.ca/ )

*[Authors' Response]: Thank you, we have reformatted the reference.*

L535 – Is this the database for the inventory (at Nordicana D?) – There should be additional information including doi number.

*[Authors' Response]: Thank you, we have reformatted the reference.*

L538-539 – Is this a conference presentation/abstract, unpublished report? Provide additional information.

*[Authors' Response]: Thank you, we have reformatted the reference.*

Figure S3 – Why only show where permafrost is not present based on 2013-17 study? It would be more useful to also include where permafrost was present during the 2013-17 study.

*[Authors' Response]: We have removed this figure from the supplemental material.*

Figure S5 – I believe Tarnocai et al. (2011) also indicates whether peatland is frozen or unfrozen. Wouldn't it be useful to show this on the map?

*[Authors' Response]: We have presented a comparison between the inventoried peatland permafrost complexes and the Tarnocai et al. (2011) perennially frozen peatlands product in the supplemental material (Supplement Sect. S4). Unfortunately, this dataset suggests that there are no perennially frozen peatlands in any part of Labrador.*

Figure S8 – How useful is this comparison given Obu et al. map is based on an equilibrium permafrost distribution and past climate conditions are not considered? Since permafrost

aggradation in this region likely occurred under a colder climate than present, the Obu et al. map will underestimate the permafrost occurrence.

*[Authors' Response]: We believe that the disagreement with Obu et al. (2019) in our region reflects performance issues with their implementation of the TTOP model rather than the utility of the TTOP model itself. Unpublished work by Way and Lewkowicz presented at the Eastern Snow Conference in 2017 showed that discrepancies between TTOP spatial models (e.g., Way and Lewkowicz, 2016) and observations of peatland permafrost along the southern coast of Labrador could largely be reconciled with an improved snow redistribution algorithm and more precise land cover maps. Further, while we agree that much of the literature suggests that peatland permafrost, especially if found near its southern limit, tends to be relict permafrost that may be in disequilibrium with the current climate, we note that one-dimensional thermal modelling for two palsas in southeastern Labrador by Way et al. (2018) found that these landforms were largely in equilibrium with current climate conditions. We have elaborated on these issues, including potential issues in the interpretation of TTOP model results, in the supplemental material (Supplement Sect. S4).*

L71-72 – Heginbottom et al. – see earlier comment

*[Authors' Response]: Thank you, we have reformatted the reference.*

L86-88 – More information about these publications should be provided. Is the NRCan Land cover map the one described below (it might also be from National Atlas 6^(th) Edition reference outline series 6409).

Canada's land cover; Latifovic, R. Natural Resources Canada, General Information Product 119e, (ed. version 2015), 2019, 1 sheet, https://doi.org/10.4095/315659

*[Authors' Response]: Thank you, we have reformatted the reference.*

L100-101 – Missing info for Tarnocai et al. This is Geological Survey of Canada Open File 6561 and has a doi number – check GEOSCAN

*[Authors' Response]: Thank you, we have reformatted the reference.*

---

## Author Comment (AC3)

**AUTHOR RESPONSES TO REFEREE 3 COMMENTARY ON MANUSCRIPT 2022-38**

Manuscript ID#: **2022-38**

Title: **Significant underestimation of peatland permafrost along the Labrador Sea coastline**

First Contact: **Yifeng Wang**

Second Contact: **Robert Way**

**REFEREE 3 (DR. STEVE KOKELJ)**

*[Authors' Response]: We thank Dr. Steve Kokelj for taking the time to provide helpful comments on our manuscript and supplemental materials. We have responded to each comment below and have made corresponding changes to the revised manuscript and supplemental materials.*

**COMMENTS TO THE AUTHORS**

The Introduction is reasonably effective at framing the study but should be further strengthened by better linking the state of knowledge with clearly articulated research questions or hypotheses. This will help to better frame the content of the paper, and provide clear logic behind the methods and analyses that are implemented.

*[Authors' Response]: We agree and have included a sentence near the end of the Introduction that more clearly outlines the main hypothesis of the study, which is that peatland permafrost landforms are abundant in some areas along the Labrador Sea coastline.*

In relation to this point, there is a fair bit of data shown in the Supplementary materials, some of which seem central to the paper, while other figures in the main manuscript host relatively small amounts of information (F1, 2a). Some figure content could be better organized to make more economic use of figure space while highlighting the data that best supports key arguments.

*[Authors' Response]: We agree and have rearranged some of the figures between the main manuscript and the supplemental material.*

Some minor editorial adjustments and additions to the figures would be helpful to more clearly define the spatial scope of the study. Early in the manuscript, it seemed that the paper developed Labrador-wide datasets, but only later in the manuscript did it become clear that the manuscript was focused on the coastal region as indicated in the title. Also, it would be useful to express whether the inventory was aimed to be exhaustive or whether it is thought to represent a subsample of the total population of the features within the focal area of study.

*[Authors' Response]: We agree and have modified Figure 1 to include an outline for the primary study area, corresponding to the area within 100 km of the Labrador Sea coastline. We have also included a Section 2.4 (Inventory extent) that describes the extent of the inventory and the justification for our focus on the coast. We have also included additional details in Section 3 (Methods) that includes mention of the sample nature of the inventory, as opposed to a full census or total population.*

I think that the paper would also benefit significantly if the point data could be more effectively linked to some spatial characteristics of the peatlands. In this regard, I suggest three points to consider. First, it would be useful to clearly express the rule-base for decisions of how and where

researchers dropped points to indicate the presence of a (permafrost) peatland complex. In Figure S9 the points seem to represent discrete features, however, it is less apparent why multiple points are dropped in peatland areas in Figure S10. In relation to this point, I think it would add significant value to the paper if the points could be attributed by a size index describing the peatland. This could be through establishing categories based on the area (discrete/small, basin/medium, landscape/large). Alternatively, or in addition, it would be useful to digitize a random subsample of peatlands to show the size distribution of a sample population. This would better contextualize the point dataset giving the inventory more "depth" and providing a better picture of the areal coverage of permafrost peatlands. This data would also provide the Authors with a solid platform for future analyses. It would be useful to include a table showing the data model describing attributes that were collected by the inventory.

*[Authors' Response]: We have provided additional information in Section 3.2.1 (Identifying wetlands of interest (WOIs)) on the criteria that were considered when identifying individual WOIs. To support this, we have generated a new figure that shows how the delineation of the WOIs was interpreted, and this figure is included in the supplemental material (Supplement Sect S1).*

*We have also provided additional information in Section 3.2.1 (Identifying wetlands of interest (WOIs)) on the approximate range in the size of WOIs. While information on the size of the surrounding peatland (discrete/small, basin/medium, landscape/large) is relevant to a wetland mapping initiative, we are worried about an implication that larger peatlands may have more permafrost area when that is not something that we can evaluate at this point. Rather than provide an estimate of the area of each WOI, we have conducted an additional analysis, in which we have classified each likely and possible peatland permafrost complex as palsa, peat plateau, or mixed (palsa and peat plateau). This information has been included in the main manuscript (Figure 6) and will also help provide us with a solid platform for future analyses, especially given the relative permafrost coverage by peat plateaus compared to palsas.*

*As suggested, we have included a table in the supplemental material (Supplement Sect. S2) that shows the attributes that were collected as part of the inventorying and validation process.*

I generally like the comparisons between the data generated by this project and the broad-scale spatial products. I think the comparisons are made in a reasonable manner, despite the difficulty of direct comparison with most of these broad-scale datasets because what they represent can be unclear. Some straightforward quantitative comparisons that show the degree of agreement between empirical permafrost peatland observations and grid cell classifications for the datasets portrayed in Figure 5 or S5 would be useful and should be added to the results section. The implications of these results can remain in the discussion.

*[Authors' Response]: We have presented quantitative comparisons that show the degree of agreement between our inventory and the datasets portrayed in Figure S1 in Section S4 of the supplemental material. We have modified this section to also include quantitative comparisons that show the degree of agreement between our inventory and four peatland permafrost distribution products. We appreciate the suggestion to include these comparisons as part of the results of the main manuscript, but we have decided to keep them in the supplemental material (Supplement Sect. S4) as our inventory represents only a sample of some of the largest peatland permafrost complexes in coastal Labrador. These comparisons may be more suitable for a future manuscript that considers area-based estimates of peatland permafrost in Labrador.*

P1 L10-13. Consider making a clear statement of the general distribution of peatlands across Labrador early in the paper to help frame this study. This added context would help a reader not familiar with the region.

*[Authors' Response]: We have provided a statement in Section 2.2 (Physical environment) describing the general distribution of wetlands in Labrador. We have also provided reference to nine wetland or peatland distribution products in the supplemental material (Supplement Sect. S1).*

L15 – I think it would be useful to briefly explain what is meant by a wetland and peatland permafrost complex. Does the area of the landform matter?

*[Authors' Response]: We have provided additional information in Section 3.2.1 (Identifying wetlands of interest (WOIs)) on the criteria that were considered when identifying individual WOIs and the range in the size of WOIs.*

L21 – It is not clear why the presence of "frost susceptible sediments" is important for peatland permafrost to form. Is it that peatlands typically develop in flat, poorly drained environments often characterized by lacustrine or glaciolacustrine deposits, which also happen to be frost susceptible?

*[Authors' Response]: We discuss the importance of sediment type, particularly of marine deposits like marine clays and silts, for peatland permafrost distribution in Section 5.1 (Distribution of peatland permafrost in Labrador). The nature of the sediments beneath the peat is very important for peatland permafrost development and persistence given that permafrost extends through the peat and into the underlying sediments. The presence of frost-susceptible materials allows for the development of segregated ice and the formation of an elevated landform that facilitates permafrost persistence (snow scouring).*

Consider that total peatland counts are not the best way to highlight the relative importance of the phenomenon over a geographical area. While the totals have value in comparing permafrost vs non-permafrost peatlands, reporting the data as a frequency density (count/unit area) is more useful to understand the relative importance of the phenomenon, it can be portrayed spatially, and it can be compared more readily with data from other regions.

*[Authors' Response]: We appreciate this suggestion and have presented the count per area, or density, of likely peatland permafrost complexes in Figure 4B. We agree that counts that are not contextualized by total study area can be difficult to compare with other datasets, but we are cautious about making broad assumptions about the area of peatland permafrost in coastal Labrador from this first inventory. This manuscript is intended to describe a first attempt at mapping peatland permafrost complexes in an understudied region where these kinds of landforms were previously believed to be largely absent. These suggestions will certainly be considered in the next steps of our overarching project, and additional work in modelling the area of peatland permafrost in coastal Labrador is already underway.*

L28. Suggested modification. Add "in the form of" palsas (peat mounds…)

*[Authors' Response]: We agree and have changed "as" to "in the form of".*

L28-29. Suggested modification for the definition of peat plateau. "variable-sized fields of frozen peat elevated above the general surface of the peatland"

*[Authors' Response]: We appreciate the suggestion and have changed the definition to "fields of frozen peat elevated above the general surface of the surrounding peatland".*

P2L44-51. This narrative is good, but it would also be useful to describe the distribution of peatlands in Labrador (and the coast) to better contextualize the study. There are some nice maps in the supplement but those don't get introduced until much later in the paper. If peatland distribution was integrated into a map earlier in the main manuscript it would help contextualize the discussion from L44-51.

*[Authors' Response]: We have provided a statement in Section 2.1 (Bioclimatic setting) describing the general distribution of wetlands in Labrador. We have also provided reference to nine wetland or peatland distribution products in the supplemental material (Supplement Sect. S1).*

P2-3L64-65. It would be useful to more clearly indicate the spatial scope of the study. It is implied in P2 L63-65 but should be clarified and shown in Figure 1.

*[Authors' Response]: We agree and have modified Figure 1 to include an outline for the study area, corresponding to the area within 100 km of the Labrador Sea coastline. We have also included a Section 2.4 entitled "Inventory extent" which describes the extent of the inventory and the justification for our focus on the coast.*

P3L66. Overall, the introduction is well-constructed and the need for research into peatland permafrost is apparent. Still, the final paragraph could be improved by clarifying the research questions or main hypotheses.

*[Authors' Response]: We agree and have included a sentence near the end of the Introduction that more clearly outlines the main hypothesis of the study, which is that peatland permafrost landforms are abundant along the Labrador Sea coastline.*

L86-93. To support this text it would be useful to show the relative proportion of different terrain types in one of the maps.

*[Authors' Response]: We agree that it would be useful to provide information on the different terrain types and surficial deposits. Unfortunately, surficial materials information for the entirety of Labrador is currently only available at the 1:1,000,000 scale, with some information at the 1:50,000 scale in scattered locations. Our ability to provide this information is unfortunately limited by the availability of surficial materials products at an appropriate scale and will not be possible until significant advances are made in this area by partner institutions or governments. We have included additional information in Section 5.2 (Implications for peatland permafrost and permafrost distribution in northeastern Canada) describing the lack of available surficial materials data at a suitable scale for all of Labrador.*

P4L95-109.

I find this section to be well-written and informative. It highlights data gaps and provides a nice context for your study. Some of this narrative could be situated in the introduction section to help establish the relevance of your work and to frame clear research questions.

*[Authors' Response]: Thank you! We have included some of this information in the Introduction.*

It would be useful to define the study area up front and show it in a figure early in the main manuscript.

*[Authors' Response]: We agree and have modified Figure 1 to include an outline for the study area, corresponding to the area within 100 km of the Labrador Sea coastline. We have also*

*included a Section 2.4 entitled "Inventory extent" which describes the extent of the inventory and the justification for our focus on the coast.*

It would be useful to elaborate on the description of Peatland permafrost complexes in the study area with reference to figures early on.

*[Authors' Response]: We have provided additional information in Section 3.2.1 (Identifying wetlands of interest (WOIs)) on the criteria that were considered when identifying individual WOIs and have generated a figure that shows how the delineation of the WOIs was interpreted and this figure is included as Figure 2 in the main manuscript.*

On P7 L149-154 you could clarify that variation in elevation was used to assess permafrost presence.

*[Authors' Response]: We agree and have clarified that evident shadows indicative of elevated landform edges relative to the surrounding peatland was used to assess permafrost presence.*

P7. Upon inspecting some of the supplementary materials the Authors should clarify what comprises a WOI, or a point. Was there a rule base that indicates how a researcher identified a discrete "complex", and when one vs. two points were dropped? For example, the identification of discrete wetlands seems clear on FS9, but the distinction is less obvious on FS10.

*[Authors' Response]: We have provided additional information in Section 3.2.1 (Identifying wetlands of interest (WOIs)) on the criteria that were considered when identifying individual WOIs and have generated a figure that shows how the delineation of the WOIs was interpreted. This figure is included as Figure 2 in the main manuscript.*

P8. With respect to utilizing the DJI Mini 2 as explained in the methods, I would caution promoting a "best practice" since Canadian regulations require maintaining a visual line of sight.

Source: https://www.gazette.gc.ca/rp-pr/p2/2019/2019-01-09/html/sor-dors11-eng.html

Visual line of sight see 901.11; also see definitions of VLOS.

*[Authors' Response]: We acknowledge the reviewer's concerns; however, since the DJI Mini 2 microdrone weighs less than 250 g, it is exempt from Transport Canada regulations regarding small remotely piloted aircrafts (250 g to 1 kg) and can legally be flown beyond visual line of sight. To avoid promoting a best practice of operating larger remotely piloted aircrafts beyond visual line of sight, we have removed this sentence from the text.*

Figure S3. It would be useful to show all of the survey points and the flight line.

*[Authors' Response]: We have included a figure in the supplemental material (Supplement Sect. S2, Figure S3) that shows the helicopter survey line and all WOIs that were validated via this method.*

3.3 Validation: It would be useful to describe the data model that guided the collection of the inventory information.

*[Authors' Response]: We have provided additional details in Section 3.3. (Validation of subset of WOI database) on the selection criteria for WOIs to be validated via field visits.*

Figure 2. Please indicate the study area that bounded the extent of the inventory. Also, please adjust the contrast of the "Not Permafrost Peatland" symbol to improve their visibility.

*[Authors' Response]: As suggested, we have included the primary inventory study area and adjusted the symbology for the "Not Peatland Permafrost" locations in Figure 3.*

P9 L208-211. Section 4.1 is very brief without much supporting analyses or graphics. Consider integrating this section with the next section.

*[Authors' Response]: We appreciate the suggestion but have decided to keep the text as is to help structure the results according to the different stages of the inventorying process.*

Supplement Sect. S3. Can the Authors indicate all of the points showing the different WOI categories?

*[Authors' Response]: Following previous suggestions, we have defined our study area more clearly as the area of Labrador and adjacent parts of Quebec that fall within 100 km of the Labrador Sea coastline. We have described in Section 2.4 (Inventory extent) our justification for our focus on the coast, so we have removed this figure from the supplemental material as it is no longer necessary.*

The data in Figure 4 is good and the descriptions are clear. Consider paired plots that normalize the distribution against available terrain within that class. Also, it would be interesting to see a plot of the distribution of peatlands without evidence of permafrost.

*[Authors' Response]: We appreciate the suggestion and have provided histograms characterizing the complexes that were classified as unlikely to contain peatland permafrost in the supplemental material (Supplement Sect. S3).*

P14 L278. Permafrost peatlands can also develop in flat sandy areas so that while ice segregation is commonly associated with peatland permafrost it is not a prerequisite. Here I would also suggest referencing the primary literature to support this point rather than a national-scale rule-based model.

*[Authors' Response]: We have reworded the sentence to clarify that it is specifically palsas and peat plateaus that form from the epigenetic development of segregated ice and have included additional references to better support this phrase.*

P15-17. Figure 5 and S5 host a large amount of spatial data and the Discussion narrative compares and contrasts this study with modeled outputs of related variables. Systematic comparisons of these data sets should be presented as results and the implications can then be addressed more qualitatively in the Discussion. The comparisons are interesting and should be expressed as a study objective given that Figures 5, and S5 present 13 maps with significant amounts of data aimed at comparing new results from this study with existing mapping data.

*[Authors' Response]: We have provided quantitative comparisons that show the degree of agreement between our inventory and the datasets portrayed in Figure 6 and Figure S1 in Section S4 of the supplemental material. We have modified this section to also include quantitative comparisons that show the degree of agreement between our inventory and four peatland permafrost distribution products, three of which are already presented in the main manuscript. We appreciate the suggestion to include these comparisons as part of the results of the main manuscript, but we have decided to keep them in the supplemental material (Supplement Sect. S4) to keep the manuscript as compact as possible and to avoid deterring the focus of the manuscript away from its intent of describing an abundance of palsas and peat plateaus in an understudied region where permafrost was previously believed to be largely absent. Further, as our inventory*

*represents only a sample of some of the largest peatland permafrost complexes in coastal Labrador, these comparisons may be more suitable for a future manuscript that considers area-based estimates of peatland permafrost in Labrador.*

To reiterate a previous point, I think it is also helpful to present results as a count per area because reporting total numbers of peatland occurrences does not provide a great sense of their spatial coverage or regional importance. Furthermore, counts that are not contextualized by total study area are difficult to compare with other datasets.

*[Authors' Response]: We appreciate this suggestion and have presented the count per area, or density, of likely peatland permafrost complexes in Figure 4B. We agree that counts that are not contextualized by total study area can be difficult to compare with other datasets, but we are cautious about making broad assumptions about the total area of peatland permafrost in coastal Labrador from this first inventory. We have made some progress towards area-based analyses and estimates of peatland permafrost coverage in Labrador, and we have included some of these steps as part of the main manuscript. For example, we have conducted and included an additional analysis, in which we have classified each likely and possible peatland permafrost complex type as palsa, peat plateau, or mixed (palsa and peat plateau). This information has been included in the main manuscript (Figure 6) and provides interesting insights into the extent of permafrost in areas that are dominated by peat plateaus versus palsas. We appreciate the reviewer's suggestions, and they will certainly be considered in the next steps of our overarching project to study the distribution and sensitivity of peatland permafrost in coastal Labrador.*

---

## Referee Report (RR1)

TC-2022-38_revision_SVK

I have reviewed the revised manuscript and the Authors' responses and commend them for robustly addressing comments. Some relatively simple adjustments have improved the manuscript, such as clarifying the spatial scope of the inventory. I note that the Authors have included some additional figures in the main body of the manuscript to better explain methods or support results and interpretations. Reviewers' requests for further detail on methods or interpretations were addressed by some elaboration or additions of text, and by adding material to the Supplement. The revised version explains the inventory methods and workflow better and provides some interpretation of variation in the landform type. The study is straightforward yet represents a good empirically-based contribution to the knowledge of permafrost distribution and periglacial landforms over an important region of northeastern Canada. The data, analyses, and synthesis are sufficiently robust to warrant publication in The Cryosphere. I expect the paper will be well cited because of its regional significance, straightforward, and clearly expressed methods, and because it highlights the importance of empirical datasets in understanding the thaw-sensitivity of Arctic Landscapes.

I have a few very minor points for the Authors to consider in preparing their manuscript.

Specific points

P3 L69. In the first review, developing hypotheses were suggested to help a reader understand the scientific focus of the paper and to frame the methods and analyses. In the revised version, I don't think that the hypothesis as stated is explicitly tested in the study, so consider either reframing it or stating it more generally as a few objectives that allow a reader to understand the logic behind what is going to be presented and how it will be analyzed. So for example, Objectives were to develop inventory methods to…..; evaluate the distribution of permafrost peatlands to……; compare the empirical data to model products to ……...

While this may be more of a point of style, I think slight improvement and additional information will help better frame a good study and help a reader understand what to expect in the manuscript.

I find the addition of Figure 2 helpful. Is there the possibility of linking an oblique shot to the imagery so that others attempting to map similar features have a point of visual reference?

Pg 16 L297-303. The addition of text describing different forms and their frequency of occurrence has been helpful.

I would suggest adding a reference to Figure 6 on P16 L297-303.

P18. L330-333. Do the Authors think that drainage may contribute to the resilience of permafrost in Labrador peatlands? I am not familiar with the terrain, however, in the poorly-drained Taiga Plains, lateral degradation due to advection contributes to the rapid expansion of collapse scars and basins. I raise this only as a point of interest given that they suggest extremely large thermal offsets.

P18. I don't follow the logic behind vertical ice lens size distribution and grain size. Slight elaboration would be helpful so the physical basis for the statement can be understood without going to Allard and Rousseau, 1999.

Figure 7A-D. Please distinguish the inventoried area.

P21 L403-404. This section reads well and provides a helpful discussion of the inventory's strengths and weaknesses.

Figure S6. Is there a scale unit missing for map A?

---

## Author Response (AR2)

**AUTHOR RESPONSES TO EDITOR COMMENTARY ON MANUSCRIPT 2022-38**

Manuscript ID#: **2022-38**

Title: **Significant underestimation of peatland permafrost along the Labrador Sea coastline**

First Contact: **Yifeng Wang**

Second Contact: **Robert Way**

**EDITOR**

*[Authors' Response]: We thank Dr. Hauck for taking the time to review the referee reports and our responses to each of the comments. We greatly appreciate their consideration of this manuscript. We have responded to the three referee reports below and have made corresponding changes to the revised manuscript. A summary of implemented changes has been included as follows:*

*SUMMARY OF ALL CHANGES:*

*[1] All technical corrections were incorporated as suggested.*

*[2] All corrections to figures were incorporated as suggested.*

*[3] We have clarified that the sediments in L336 are frost-susceptible sediments.*

*[4] We have clarified the discussion on the thickness of ice lenses that may develop in fine- versus coarse-grained sediments.*

**AUTHOR RESPONSES TO REFEREE 1 COMMENTARY ON MANUSCRIPT 2022-38**

Manuscript ID#: **2022-38**

Title: **Significant underestimation of peatland permafrost along the Labrador Sea coastline**

First Contact: **Yifeng Wang**

Second Contact: **Robert Way**

**REFEREE 1**

*[Authors' Response]: We thank Referee 1 for taking the time to provide helpful comments and revisions on our manuscript. We appreciate their constructive review, which has certainly helped to improve the manuscript. We have responded to each comment below and have made corresponding changes to the revised manuscript. A summary of implemented changes has been included at the bottom of this response.*

**COMMENTS TO THE AUTHORS**

Ln 94 consider changing "solid precipitation" to "snow"

*[Authors' Response]: We agree and have changed "solid precipitation" to "snow".*

Ln 107 change to "kyr BP"

*[Authors' Response]: We agree and have changed "k years BP" to "kyr BP".*

Ln 113 "flat deposits" or "flat surfaces" – ie only over sedimentary deposits or over bedrock as well?

*[Authors' Response]: We agree and have changed "flat deposits" to "flat areas".*

Ln 124 change "does become" to "is"

*[Authors' Response]: We agree and have changed "does become" to "is".*

Ln 188 change to "quality-control check"

*[Authors' Response]: We agree and have added a hyphen to change "quality control check" to "quality-control check".*

Ln 241 change "possibly" to "possible". Although, in text, the term "possibly" is used, it is best to best consistent with terms used in final mapping and as relating to Figures 4 and 5.

*[Authors' Response]: We agree and have changed "possibly" to "possible".*

Ln 245 "limited to the largest" – suggest adding in brackets, an indication of what minimum surface area this represents. Example (≥ ## m2). At present it is unclear what is meant by the largest.

*[Authors' Response]: We have changed the sentence to clarify the minimum size of the peatland permafrost landforms that were included in the inventory. This is also mentioned in Section 3.2.1 Identifying wetlands of interest (WOIs).*

Ln 288 change to "possible"

*[Authors' Response]: We agree and have changed "possibly" to "possible".*

*SUMMARY OF CHANGES:*

*[1] All technical corrections were incorporated as suggested.*

**AUTHOR RESPONSES TO REFEREE 2 COMMENTARY ON MANUSCRIPT 2022-38**

Manuscript ID#: **2022-38**

Title: **Significant underestimation of peatland permafrost along the Labrador Sea coastline**

First Contact: **Yifeng Wang**

Second Contact: **Robert Way**

**REFEREE 2**

The detailed response by the authors to the review comments is very much appreciated. Revisions have been made to the manuscript to address the review comments, such as including additional material and clarifications (e.g. clear definition of study area). These revisions have improved the MS and in my view it is acceptable for publication with a few very minor revisions (mostly editorial) as outlined below. I look forward to seeing the published paper.

*[Authors' Response]: We thank Referee 2 for taking the time to provide helpful comments and revisions on our manuscript. We appreciate their constructive review, which has certainly helped to improve the manuscript. We have responded to each comment below and have made corresponding changes to the revised manuscript. A summary of implemented changes has been included at the bottom of this response.*

**COMMENTS TO THE AUTHORS**

L37 – "activities" could probably be deleted

*[Authors' Response]: We agree and have removed "activities".*

L51 – Suggested revision: "….peatland permafrost occurrence has…"

*[Authors' Response]: We agree and have added "occurrence" to the sentence.*

L327 – Do you mean permafrost persistence rather than landform persistence?

*[Authors' Response]: We agree and have changed "landform" to "permafrost".*

L332 – "support" might be a better word than "protect"

*[Authors' Response]: We agree and have changed "protect" to "support".*

L336 – Are you referring specifically fine-grained sediments here?

*[Authors' Response]: We have clarified that we are referring specifically to frost-susceptible sediments, or sediments capable of facilitating frost heave. This is based on the findings of Allard*

*and Rousseau (1999), who compare deposits of clayey silt with deposits of sandy-silty clay in the formation of palsas versus peat plateaus.*

L 380-381 – Heginbottom et al. (1997) refers to a conference paper rather than the circumpolar permafrost map itself the reference of which is Brown et al. (1997): Brown J, Ferrians Jr. OJ, Heginbottom JA, Melnikov ES (1997) Circum-Arctic map of permafrost and ground-ice conditions. U.S. Department of the Interior, U.S. Geological Survey, Map CP-45 The same comment applies to the reference to Heginbottom et al. (1997) in the Supplementary Information. You could just cite Heginbottom et al. (1995) since you are referring specifically to the Permafrost Map of Canada.

*[Authors' Response]: We have changed the "Heginbottom et al. (1997)" reference in L380-381 to "Heginbottom et al. (1995)". We have changed the "Heginbottom et al. (1997)" reference in the Supplementary Information to "Brown et al. (1997)".*

L511-512 – You should give page numbers for the paper

*[Authors' Response]: We have revised the reference and included the page numbers for the paper instead of the number of pages of the paper.*

L524 – The URL should be provided for the website

*[Authors' Response]: We have revised the reference and included the URL for the Canadian Climate Normals website.*

Section S5 L58 – This should be "larger scale permafrost distribution products" since the various national and circumpolar scale maps mentioned are smaller scale products.

*[Authors' Response]: We agree and have changed "smaller" to "larger".*

*SUMMARY OF CHANGES:*

*[1] All technical corrections were incorporated as suggested.*

*[2] We have clarified that the sediments in L336 are frost-susceptible sediments.*

**AUTHOR RESPONSES TO REFEREE 3 COMMENTARY ON MANUSCRIPT 2022-38**

Manuscript ID#: **2022-38**

Title: **Significant underestimation of peatland permafrost along the Labrador Sea coastline**

First Contact: **Yifeng Wang**

Second Contact: **Robert Way**

**REFEREE 3 (DR. STEVE KOKELJ)**

I have reviewed the revised manuscript and the Authors' responses and commend them for robustly addressing comments. Some relatively simple adjustments have improved the manuscript, such as clarifying the spatial scope of the inventory. I note that the Authors have included some additional figures in the main body of the manuscript to better explain methods or support results and interpretations. Reviewers' requests for further detail on methods or interpretations were addressed by some elaboration or additions of text, and by adding material to the Supplement. The revised version explains the inventory methods and workflow better and provides some interpretation of variation in the landform type. The study is straightforward yet represents a good empirically-based contribution to the knowledge of permafrost distribution and periglacial landforms over an important region of northeastern Canada. The data, analyses, and synthesis are sufficiently robust to warrant publication in The Cryosphere. I expect the paper will be well cited because of its regional significance, straightforward, and clearly expressed methods, and because it highlights the importance of empirical datasets in understanding the thaw-sensitivity of Arctic Landscapes.

*[Authors' Response]: We thank Dr. Steve Kokelj for taking the time to provide helpful comments and revisions on our manuscript. We appreciate their constructive review, which has certainly helped to improve the manuscript. We have responded to each comment below and have made corresponding changes to the revised manuscript. A summary of implemented changes has been included at the bottom of this response.*

**COMMENTS TO THE AUTHORS**

P3 L69. In the first review, developing hypotheses were suggested to help a reader understand the scientific focus of the paper and to frame the methods and analyses. In the revised version, I don't think that the hypothesis as stated is explicitly tested in the study, so consider either reframing it or stating it more generally as a few objectives that allow a reader to understand the logic behind what is going to be presented and how it will be analyzed. So for example, Objectives were to develop inventory methods to…..; evaluate the distribution of permafrost peatlands to……; compare the empirical data to model products to ……... While this may be more of a point of style, I think slight improvement and additional information will help better frame a good study and help a reader understand what to expect in the manuscript.

*[Authors' Response]: We have reformatted the last paragraph of the Introduction as suggested and have outlined three objectives:*

1) *To use a multi-stage, consensus-based review process, coupled with extensive validation efforts from a combination of field visits and low-altitude image and video acquisitions, to develop a point inventory of contemporary peatland permafrost complexes in coastal Labrador*
2) *To characterize the distribution of peatland permafrost in coastal Labrador using selected climatic and physiographic variables*
3) *To provide insights into the reliability of relevant peatland permafrost and permafrost distribution products, which currently claim an absence or low abundance of both peatland permafrost and permafrost along the Labrador Sea coastline*

I find the addition of Figure 2 helpful. Is there the possibility of linking an oblique shot to the imagery so that others attempting to map similar features have a point of visual reference?

*[Authors' Response]: We agree and have revised the figure by including oblique shots of two example areas to provide nadir and oblique perspectives of the same wetlands of interest and by removing two of the original example areas to avoid overcrowding the figure.*

Pg 16 L297-303. The addition of text describing different forms and their frequency of occurrence has been helpful.

*[Authors' Response]: Thank you.*

I would suggest adding a reference to Figure 6 on P16 L297-303.

*[Authors' Response]: We agree and have added a reference to Figure 6 on P16 L298.*

P18. L330-333. Do the Authors think that drainage may contribute to the resilience of permafrost in Labrador peatlands? I am not familiar with the terrain, however, in the poorly-drained Taiga Plains, lateral degradation due to advection contributes to the rapid expansion of collapse scars and basins. I raise this only as a point of interest given that they suggest extremely large thermal offsets.

*[Authors' Response]: We do think that drainage plays an important role in permafrost resilience. We recognize that we do not discuss drainage in particular detail in this study, as it focuses primarily on identifying contemporary peatland permafrost complexes rather than identifying past peatland permafrost landforms through mapping of collapse scars or thermokarst ponds. However, our research group is currently working to investigate historical changes in the lateral extent of palsas and peat plateaus in selected complexes, where drainage is expected to play an important role in long-term changes. Thermal modelling may also help to address questions related to the importance of drainage and heat transfer through water flow in palsas and peat plateaus.*

P18. I don't follow the logic behind vertical ice lens size distribution and grain size. Slight elaboration would be helpful so the physical basis for the statement can be understood without going to Allard and Rousseau, 1999.

*[Authors' Response]: We have revised the sentence to focus more on differences in the thickness of ice lenses between finer versus coarser sediment types.*

Figure 7A-D. Please distinguish the inventoried area.

*[Authors' Response]: We have revised the figure and included a black boundary for the study area in each of the four maps to distinguish the inventoried area.*

P21 L403-404. This section reads well and provides a helpful discussion of the inventory's strengths and weaknesses.

*[Authors' Response]: Thank you.*

Figure S6. Is there a scale unit missing for map A?

*[Authors' Response]: We have added the units of metres as "(m)" to the legend for map A in Figure S6.*

*SUMMARY OF CHANGES:*

*[1] All technical corrections were incorporated as suggested.*

*[2] All corrections to figures were incorporated as suggested.*

*[3] We have clarified the discussion on the thickness of ice lenses that may develop in fine- versus coarse-grained sediments.*

---

## Author Response (AR3)

**AUTHOR RESPONSES TO EDITOR COMMENTARY ON MANUSCRIPT 2022-38**

Manuscript ID#: **2022-38**

Title: **Significant underestimation of peatland permafrost along the Labrador Sea coastline**

First Contact: **Yifeng Wang**

Second Contact: **Robert Way**

**EDITOR**

thank you very much for your revised version and your detailed author comments. The manuscript further improved substantially and you responded very well to all comments by the reviewers. Due to the thorough reviews and your careful revisions I have nothing to add at this stage and am happy to state that the paper can now be accepted for publication.

*[Authors' Response]: We thank Dr. Hauck very much for his support throughout this process.*

The only recommendation at this stage from my side would be to reduce the size of the figure sub-panel headers (a), (b) etc within the figure subpanels - especially in Fig. 2, but also in many of the other figures. The size of the headers (a), (b) in Fig. S1 seems to be more appropriate than in Figures 1-4 and 6-7 for example.

*[Authors' Response]: As suggested, we have reduced the size of the figure sub-panel headers in Figures 1, 2, 3, 4, 5, 6, and 7 in the manuscript and Figures S4, S5, and S6 in the supplemental.*

*SUMMARY OF ALL CHANGES:*

*[1] Reduced the size of the figure sub-panel headers.*